# Wbp2 is required for normal glutamatergic synapses in the cochlea and is crucial for hearing

Annalisa Buniello[1,2,*], Neil J Ingham[1,2], Morag A Lewis[1,2], Andreea C Huma[2], Raquel Martinez-Vega[2,3,4], Isabel Varela-Nieto[3,4], Gema Vizcay-Barrena[5], Roland A Fleck[5], Oliver Houston[6], Tanaya Bardhan[6], Stuart L Johnson[6], Jacqueline K White[2], Huijun Yuan[7], Walter Marcotti[6] & Karen P Steel[1,2,**]

## Abstract

*WBP2* encodes the WW domain-binding protein 2 that acts as a transcriptional coactivator for estrogen receptor α (*ESR1*) and progesterone receptor (*PGR*). We reported that the loss of *Wbp2* expression leads to progressive high-frequency hearing loss in mouse, as well as in two deaf children, each carrying two different variants in the *WBP2* gene. The earliest abnormality we detect in Wbp2-deficient mice is a primary defect at inner hair cell afferent synapses. This study defines a new gene involved in the molecular pathway linking hearing impairment to hormonal signalling and provides new therapeutic targets.

**Keywords** glutamate excitotoxicity; hearing impairment; hormonal signalling; ribbon synapses; transcriptional coactivator

**Subject Categories** Genetics, Gene Therapy & Genetic Disease; Neuroscience

## Introduction

Progressive hearing loss is very common in the human population, but we know very little about its molecular basis and have no medical therapies. One subtype of cochlear pathology is thought to involve primary degeneration of afferent neurons of the cochlea (Schuknecht, 1993; Sergeyenko *et al*, 2013). In the auditory system, sensory hair cells are responsible for transforming an acoustic, mechanical stimulus into an electrical signal via the activation of mechanically gated ion channels and the generation of a receptor potential (Fettiplace & Kim, 2014). The inner hair cells (IHCs), which are the primary receptors of the mammalian cochlea, relay acoustic signals with remarkable acuity and temporal precision to

the brain through the release of glutamate to their afferent neurons (Fuchs, 2005). IHCs have a specialised synapse with a pre-synaptic ribbon thought to gather synaptic vesicles to facilitate rapid release upon hair cell depolarisation. These ribbon synapses are innervated by the unbranched dendrites of spiral ganglion neurons, with a single ribbon synapse signalling to each afferent fibre, highlighting the importance of accurate neurotransmission at these synapses (Fuchs, 2005). The outer hair cells (OHCs) serve both as sensory receptors and as biological amplifiers. Their sensory function is less well understood because their afferent innervation has small unmyelinated axons and constitutes only 5% of the cochlear nerve (Thiers *et al*, 2008). The molecular pathways underlying hair cell synaptic development and plasticity are still poorly understood.

Steroid hormones are known to be implicated in normal auditory function (Hultcrantz *et al*, 2006; Charitidi *et al*, 2009, 2010). For example, mutations of several genes involved in oestrogen signalling lead to hearing impairment, such as *Esr2*, *Esrrb* and *Esrrg* in mice and *ESRRB* in human deafness type DFNB35, and hearing impairment is a feature of oestrogen deficiency in Turner syndrome in humans (Chen & Nathans, 2007; Collin *et al*, 2008; Meltser *et al*, 2008; Hederstierna *et al*, 2009; Simonoska *et al*, 2009; Nolan *et al*, 2013). Furthermore, oestrogen signalling protects against noise-induced hearing loss (Meltser *et al*, 2008). A key role for oestrogen-dependent signalling has been demonstrated in other neural cells, specifically affecting synapses (Singer *et al*, 1996; McEwen *et al*, 2001; Akama & McEwen, 2003; Liu *et al*, 2008; Baudry *et al*, 2013). However, the wide range of other effects of oestrogen has meant that oestrogen-based therapies are not generally considered for hearing loss, and trials of generic hormone replacement therapies have had mixed outcomes for hearing (Kim *et al*, 2002; Caruso *et al*, 2003; Kilicdag *et al*, 2004; Guimaraes *et al*, 2006). Clinical trials may have a better chance of success if carried out on stratified subsets of hearing-impaired people with an underlying pathology that is likely to involve oestrogen signalling, so an improved

1  Wolfson Centre For Age-Related Diseases, King's College London, London, UK
2  Wellcome Trust Sanger Institute, Wellcome Trust Genome Campus, Hinxton, UK
3  Instituto de Investigaciones Biomédicas Alberto Sols, CSIC-UAM, Madrid, Spain
4  Centre for Biomedical Network Research on Rare Diseases (CIBERER), Unit 761, Instituto de Salud Carlos III, Madrid, Spain
5  Centre for Ultrastructural Imaging, King's College London, London, UK
6  Department of Biomedical Science, University of Sheffield, Sheffield, UK
7  Medical Genetics Center, Southwest Hospital, Third Military Medical University, Chongqing, China
   *Corresponding author. Tel: +44 207 848 6803; E-mail: annalisa.buniello@kcl.ac.uk
   **Corresponding author. Tel: +44 207 848 6203; E-mail: karen.steel@kcl.ac.uk

understanding of the role of oestrogen in hearing will contribute to stratification strategies. In addition, manipulation of oestrogen signalling may be more effective as a treatment for hearing loss if targeted to the auditory system, so understanding any oestrogen-sensitive molecular networks specific to hearing may open up new targets for therapies for hearing loss.

In order to investigate the functional link between hormonal signalling and hearing impairment and identify new targets for therapies, we used Wbp2-deficient mice as a genetic tool. *WBP2* encodes the WW domain-binding protein 2, which is phosphorylated before translocating into the nucleus where it acts as a transcriptional coactivator for the estrogen and progesterone receptors *ESR1* and *PGR* (Dhananjayan *et al*, 2006; Lim *et al*, 2011). It plays a role in regulating the expression of target genes via hormone-dependent interaction (Dhananjayan *et al*, 2006). We report here our identification of a role for *Wbp2* in hearing through our high-throughput screen for hearing impairment using auditory brainstem responses (ABRs) of newly generated mouse mutants (White *et al*, 2013). The loss of Wbp2 causes progressive high-frequency hearing loss from as early as 4 weeks of age in mouse. We demonstrate that this is associated with reduced expression of *Esr1*, *Esr2* and *Pgr* in the cochlea and disruption of expression of key post-synaptic proteins. Moreover, we report the cases of a 5-year-old boy and a 9-year-old girl with severe to profound prelingual deafness, each carrying two point variants in heterozygosis in the *WBP2* gene.

## Results

*Wbp2^tm2a(EUCOMM)Wtsi*-mutant mice were generated and maintained on a C57BL/6N genetic background (Skarnes *et al*, 2011; White *et al*, 2013) (Fig 1A). Western blot analysis showed the absence of both Wbp2 isoforms in the brain of homozygous mutants at P28 (Fig 1B).

Quantitative real-time PCR showed the knockdown of transcription to < 1% of normal mRNA levels both in the cochlea (0.5%) and in the eye (0.8%) (Fig 1C). Despite reduced levels of *Wbp2* mRNA in the cochlea of heterozygotes (44%), they did not show any auditory phenotype (see below). X-Gal staining at post-natal day (P) 14 showed the expression of *Wbp2* in all major cochlear structures and very strongly in the cell bodies of spiral ganglion neurons (Fig 1D–F).

### Wbp2-deficient mice show progressive high-frequency hearing loss, but no other defects

Auditory function in *Wbp2*-mutant mice was investigated using auditory brainstem responses (ABR), reflecting the activity of both afferent and efferent neurons. At P14, ABR thresholds in mutants were comparable to those in littermate controls (Fig 2A). Loss of sensitivity to frequencies of 24 kHz and above was apparent by 4 weeks (Fig 2B), and even more evident at 14 and 28 weeks (Fig 2C and D), spreading to lower frequencies by 44 weeks in mutants (Fig 2E and F).

ABR waveforms were averaged for broadband clicks (Fig 2G) and 12- and 24-kHz stimuli from 4-week-old mutants and wild-type (wt) littermates in 5-dB steps and adjusted for sensation level (SL;

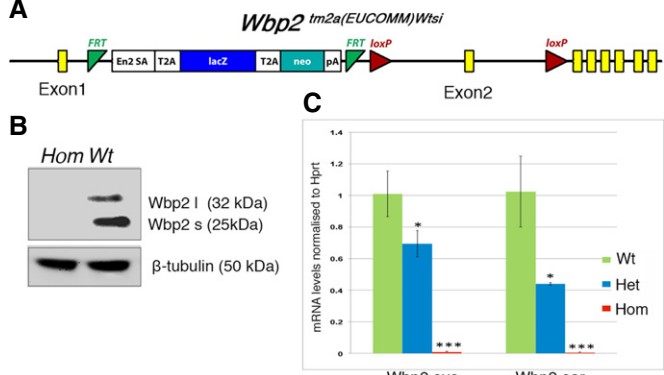

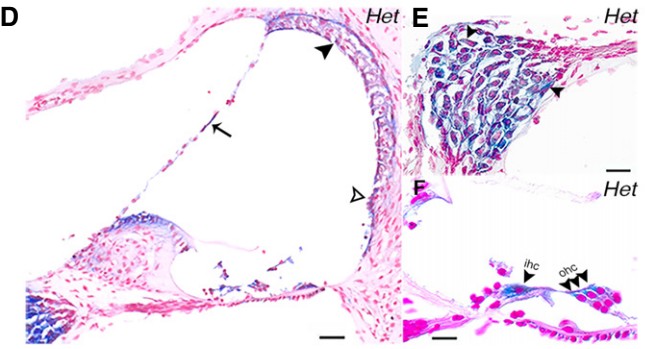

**Figure 1. Wbp2 mutation and Wbp2 expression in the cochlea.**

A   Diagram showing the design of the mutated *Wbp2* allele. A promoterless cassette including *LacZ* and *neo* genes was inserted in the second intron of the *Wbp2* gene flanked by FRT sites (green triangles). *LoxP* sites (red triangles) flank the critical exon (exon 2) of the *Wbp2* gene (exons in yellow).

B   Western blot showing no detectable Wbp2 protein in 4-week-old mutant brain compared to wt littermate controls; 5 μg of the protein lysate was subjected to 10% SDS–PAGE. β-tubulin was used as a loading control. Wbp2 l and Wbp2 s refer to the long and short isoforms, respectively.

C   Quantitative real-time PCR showing severe knockdown of *Wbp2* transcription in 4-week-old mutant (*n* = 3) inner ears and eyes, compared to wt littermate controls (*n* = 3). Heterozygotes show intermediate levels. *Hprt* was used as a control and levels are normalised to wt levels. Data plotted as mean ± SD. Two-tailed *t*-test: Wbp2 ear: het *$P$ = 0.03, hom ***$P$ = 0.000000033; Wbp2 eye: het *$P$ = 0.01, hom ***$P$ = 5.35537E-13.

D–F   X-gal staining of *Wbp2* hets at P14 showing *Wbp2* expression (blue) in all the main cochlear structures: the stria vascularis (black arrowhead in D), spiral prominence (empty arrowhead in D), Reissner's membrane (arrow in D), strong expression in the spiral ganglion cells (arrowheads in E) and in IHCs and OHCs in the organ of Corti (arrowheads in F). Scale bars: (D), 50 μm; (E, F), 20 μm. ihc: inner hair cells; ohc: outer hair cells. No X-gal staining is observed in wt controls (not shown). The X-gal reaction is always cytoplasmic.

Source data are available online for this figure.

level above threshold). While the ABRs of mutants had a similar shape compared to controls (Fig 2G), the amplitude was smaller. We quantified this reduced amplitude for the summating potential (SP) and ABR wave 1. The SP is a small positive voltage deflection before wave 1 reflecting depolarisation of IHCs (Harvey & Steel, 1992; Sergeyenko *et al*, 2013) (Fig 2G). The mutant SP was reduced compared to controls and grew at a reduced rate as stimulus level

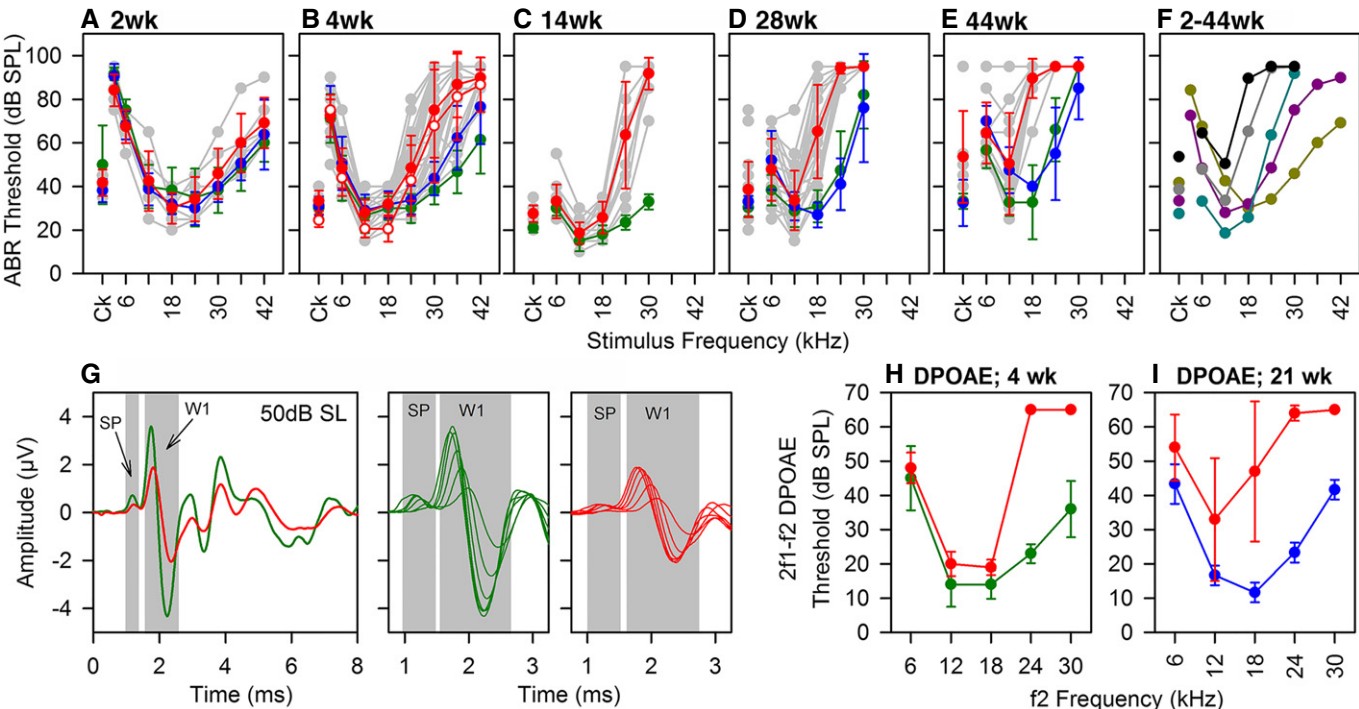

**Figure 2. Auditory responses of Wbp2-deficient mice.**

A–E  Mean ABR thresholds (± SD) for clicks and tone pips are plotted for wt (green), het (blue) and hom (red) mice at ages (A) P14 (wt, *n* = 3; het, *n* = 8; hom, *n* = 6); (B) 4 weeks (wt, *n* = 38; het, *n* = 26; hom, *n* = 37); (C) 14 weeks (wt, *n* = 10; hom, *n* = 14); (D) 28 weeks (wt, *n* = 15; het, *n* = 5; hom, *n* = 25); (E) 44 weeks (wt, *n* = 9; het, *n* = 2; hom, *n* = 11). Grey symbols and lines indicate thresholds of individual mutants. In (B), open symbols represent thresholds under urethane anaesthesia (see Materials and Methods), showing no difference compared with ketamine/xylazine used for all other thresholds.

F  Mean thresholds for mutants aged 2 weeks (yellow), 4 weeks (purple), 14 weeks (cyan), 28 weeks (grey) and 44 weeks (black).

G  Averaged click-evoked ABR waveforms from 4-week-old wt (*n* = 23, green) and mutants (*n* = 34, red), at 50-dB sensation level (SL) (left panel). SP and ABR wave 1 (W1) are indicated by grey areas. Expanded averaged SP and ABR W1 waveform patterns for 10- to 60-dB SL in 10-dB increments are plotted in green (wt; middle panel) and red (mutants; right panel), to illustrate the growth of SP and W1 with stimulus level.

H, I  Mean 2f1-f2 DPOAE thresholds (± SD) are plotted for wt (green), heterozygote (blue) and homozygous (red) mice aged 4 weeks (H: wt, *n* = 5; hom, *n* = 5) or 21 weeks (I: het, *n* = 3; hom, *n* = 5), as a function of f2 frequency.

increased, although latency was no different to controls (Fig EV1A–C). ABR wave 1 amplitudes also showed a significant reduction and a longer latency in mutants (Fig EV1D–I).

Distortion product otoacoustic emissions (DPOAEs) were recorded and showed raised thresholds, similar to that seen in ABR measurements (Fig 2H and I). In mice aged 4 weeks old, 2f1-f2 DPOAE thresholds for 6- to 18-kHz f2 tones were comparable in mutants and littermate controls, but were elevated for 24- and 30-kHz f2 tones. Impairment of DPOAEs was progressive. In 21-week-old mutants, 2f1-f2 DPOAE thresholds for all test frequencies were elevated, particularly at 18–30 kHz.

The increased ABR and DPOAE thresholds and reduced ABR amplitudes were not due to endocochlear potential deficits because these were normal in the mutants (Fig EV1J). Furthermore, the gross structure of middle and inner ears was normal and there was no obvious degeneration of hair cells or spiral ganglion neurons even up to 30 weeks old in mutants (Figs EV2 and EV3A).

*Wbp2* mutants were fertile and showed no other abnormalities in a broad phenotypic screen, other than reduced circulating amylase levels in females only, which likely has no biological significance (White *et al*, 2013).

**Compound heterozygous variants in *WBP2* identified in human deafness**

Following the discovery of *Wbp2* as a new gene involved in deafness in the mouse, the human *WBP2* gene was included in a screen of 8,087 deaf probands and 1,823 unaffected controls in China using targeted next-generation sequencing. We found variants in the *WBP2* gene in two hearing-impaired children with no other reported features. The first case was a 5-year-old boy with bilateral prelingual deafness. He was a compound heterozygote for two missense variants, c.671C>T; p.(Ala224Val) (exon 7) and c.478G>A: p.(Ala160Thr) (exon 5). The second proband was a 9-year-old girl with severe to profound bilateral hearing loss (Fig 3A). She was a compound heterozygote for two missense variants, c.487A>C; p.(Met163Leu) (exon 5) and c.478G>A:p.Ala160Thr (exon 5, in common with the first case). While audiological data from the first case were not available, data from pure tone audiometry for the second proband are shown in Fig 3A. In both families, the parents were not affected, and each parent was a heterozygous carrier for one of the variants (Fig 3B). The results were confirmed by Sanger sequencing (Fig 3B). The affected children did not carry pathogenic

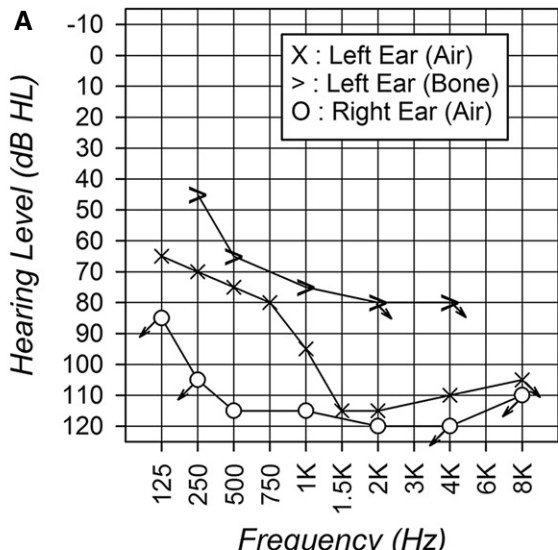

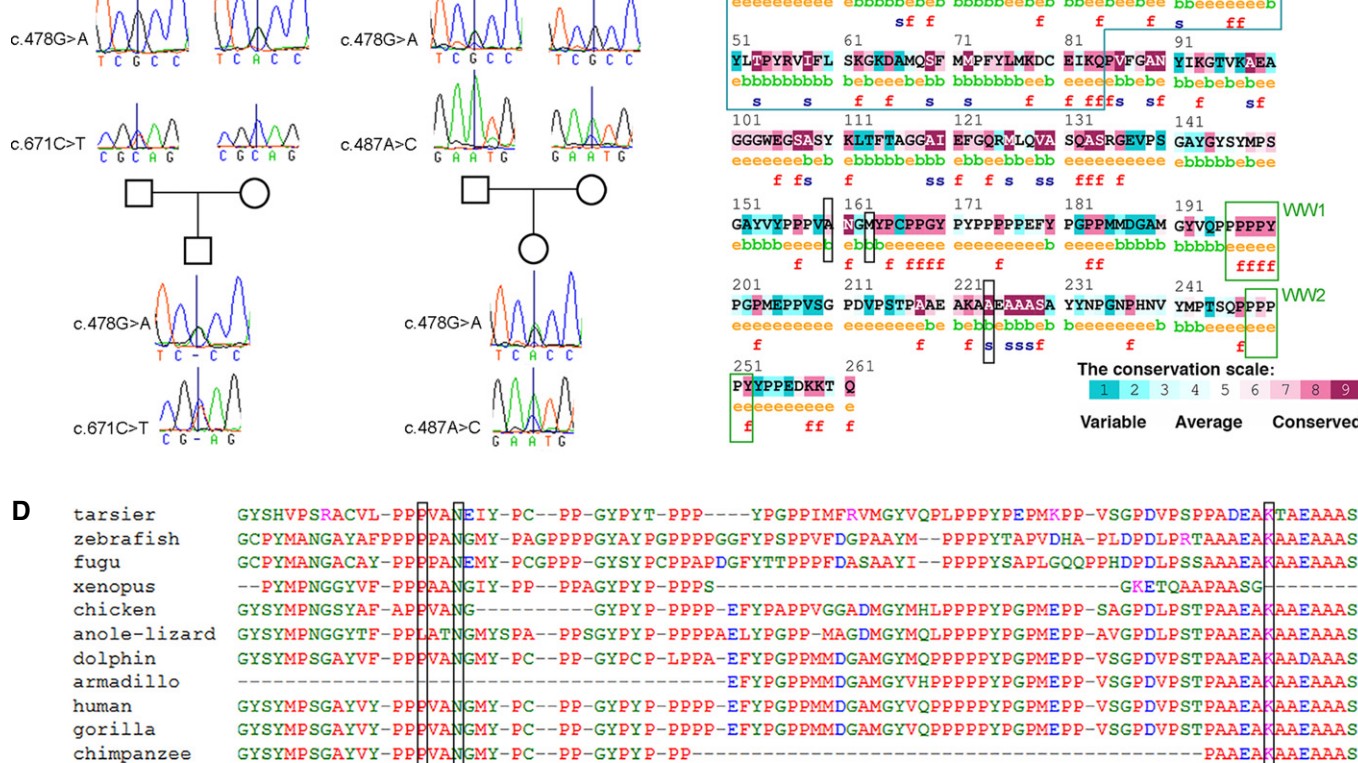

**Figure 3.  Analysis of the human variants in *WBP2*.**

A   Pure tone audiogram recorded from one of the probands, showing severe to profound bilateral asymmetric hearing loss.

B   Capillary sequence traces from fathers, mothers and probands, showing the heterozygosity of the parents for the two separate variants and the compound heterozygosity for each proband.

C   ConSeq (http://conseq.tau.ac.il/) analysis of the residues; the locations of the three human variants (p.Ala160Thr, p.Met163Leu and p.Ala224Val; boxed in black) are average to highly conserved. "b" and "e" indicate buried and exposed residues (according to the neural network algorithm), and "f" and "s" indicate predicted functional and predicted structural residues. The GRAM and WW binding domains (WW1 and WW2) are marked by blue and green boxes. The GRAM domain is thought to be an intracellular protein- or lipid-binding signalling domain and may play an important role in membrane-associated processes.

D   Alignment of the protein sequence from a range of vertebrates. The human variants are boxed in black.

variants in any of the genes known to underlie non-syndromic deafness.

We used six different predictors to assess the effects of the three variants on protein function (Table 1). The results varied from mostly tolerated to mostly damaging; none of the SNPs were universally predicted to be either tolerated or damaging. The p.Ala160Thr variant is a known SNP, rs202022024, and is very rare; in the 1,088 genomes sampled in the 1000 Genomes project, only one had this SNP. The p.Ala224Val and p.Met163Leu variants are novel. The amino acids involved in the p.Ala224Val and p.Ala160Thr changes are well conserved across species (Fig 3C and D), but the p.Met163Leu variant has an average conservation score (Fig 3C). There are 10 protein-coding splice variants of WBP2 (www.ensembl.org, accessed July 2014). The Ala160 and Met163 residues are in all but one (transcript 3), and the Ala224 residue is in seven of the splice variants, but not in transcripts 9, 10 or 11 (Fig 4A).

### The long isoform of Wbp2 is predominant in the organ of Corti

Two of the three human variants are in exon 5 (p.Met163Leu and p.Ala160Thr), which is not present in one isoform of both the human and the mouse gene. We next investigated whether, in the mouse, the long isoform containing exon 5 is particularly abundant in the inner ear. Only two Wbp2 isoforms are predicted to be present in mouse (www.ensembl.org, accessed July 2015), with the longer one containing exon 5 and the shorter one without exon 5. We sequenced Wbp2 cDNA from wild-type brain and inner ear at P28, adult organ of Corti and P4 organ of Corti and ran the cDNA on a gel before sequencing (Fig 4B and C). We found two isoforms present in the brain cDNA. Sequence traces typical of alternative splicing were observed, one including exon 5 and one excluding it. In the inner ear at P28, the long isoform could still be detected, together with a very faint band from the shorter isoform (Fig 4B and C). Only the long isoform, which includes exon 5, was present in the P4 and adult organ of Corti. Although an extremely faint small band could still be observed on the gel (Fig 4B), we could not get any sequencing trace representing the short isoform in the organ of Corti at any stage (Fig 4C). We also carried out Western blots on lysate

from wild-type brain and cochlea at P28 and again found evidence of two isoforms in the brain, with higher expression of the shorter isoform (see Fig 1B). In accordance with the sequencing data, in the cochlea at P28 the long isoform looks by far the most abundant, with a very faint band for the short isoform that can only be observed when high quantities of protein lysate are analysed on the gel (Fig 4D). These results suggested that the organ of Corti predominantly uses the long isoform of Wbp2 containing exon 5 and indicated that this isoform could be crucial for the function of WBP2 in hearing.

### Wbp2 is not required for IHC function

As we found no evidence of hair cell degeneration in the mouse Wbp2 mutants, we next investigated the biophysical properties of IHCs in isolated organ of Corti preparations from young adult mouse mutants and littermate controls. Resting membrane potentials and size of $K^+$ currents recorded from high-frequency adult mutant IHCs (P25-P33) were similar to those in littermate controls (Table EV1). Displacement of the IHC stereociliary bundle (Corns et al, 2014) of Wbp2 mutants elicited a large mechanoelectrical transducer current (1,282 ± 93pA, n = 5) similar to that of littermate controls (1,315 ± 54pA, n = 11) (Fig EV2B and C). The fraction of the mechanoelectrical transducer current activated in the resting, unstimulated bundle was also not significantly different between the two genotypes (control: 4.0 ± 0.7%, n = 11; mutant: 2.7 ± 0.5%, n = 5; values at −121 mV).

We then investigated the pre-synaptic function of IHCs in the mutant mice by measuring the increase in cell membrane capacitance ($\Delta C_m$), normally interpreted as a sign of exocytosis or neurotransmitter release. Exocytosis in high-frequency IHCs from young adult mice (P19–P33) was elicited by depolarising voltage steps from the holding potential of −81 mV. The maximal size of the $Ca^{2+}$ current ($I_{Ca}$) and the corresponding $\Delta C_m$ were similar in mutants and controls (Fig EV2D and E). The rate of neurotransmitter release was studied by measuring $\Delta C_m$ in response to depolarising voltage steps from −81 mV to −11 mV of varying duration (2 ms to 2 s), which allowed us to investigate the emptying of different synaptic vesicle pool populations (Fig EV2F and G). While relatively short stimuli (up to 50 ms) reveal the number of vesicles docked at the active zones (readily releasable pool, RRP: Fig EV2F), longer steps induce the release of vesicles from a secondarily releasable pool (SRP: Fig EV2G) that is located further away from the $Ca^{2+}$ channels (Johnson et al, 2008, 2010). The sizes of the RRP and SRP were similar between control (Fig EV2F) and mutant (Fig EV2G) IHCs. These findings indicated that Wbp2 is not required for normal pre-synaptic IHC function in young adults.

### Wbp2-deficient mice show swelling of afferent terminals and abnormal expression of AMPA receptor subunits at post-synaptic densities

To look for structural correlates of hearing impairment in the Wbp2-deficient cochlea, we next performed confocal imaging of the mouse sensory epithelium using antibodies to CtBP2 to label pre-synaptic ribbons (Kujawa & Liberman, 2009), to GluR2/GluR3 AMPA subunits to label post-synaptic densities (PSDs) (Collingridge et al, 2009) and to neurofilament to label unmyelinated nerve fibres (Kujawa &

**Table 1. The different predictions of the effects of the three human variants on WBP2 protein by six mutation predictors.**

|  | p.Ala224Val | p.Ala160Thr | p.Met163Leu |
|---|---|---|---|
| Fathmm | Tolerated | Tolerated | Tolerated |
| Mutation Assessor | **Medium impact** | Low impact | Low impact |
| Mutation Taster | **Disease causing** | **Disease causing** | **Disease causing** |
| SIFT (manual) | **Affect protein function** | Tolerated | Tolerated |
| SIFT (automated) | Tolerated | Tolerated | Tolerated |
| SIFT (ensembl) | **Deleterious** | Tolerated | Tolerated |
| PolyPhen2 | **Possibly damaging** | Benign | Benign |
| PhD-SNP | Neutral | Neutral | Neutral |

The text in the table has been bolded to clearly distinguish predictions that the mutation would be deleterious (bold) from predictions that it would be tolerated (normal text).

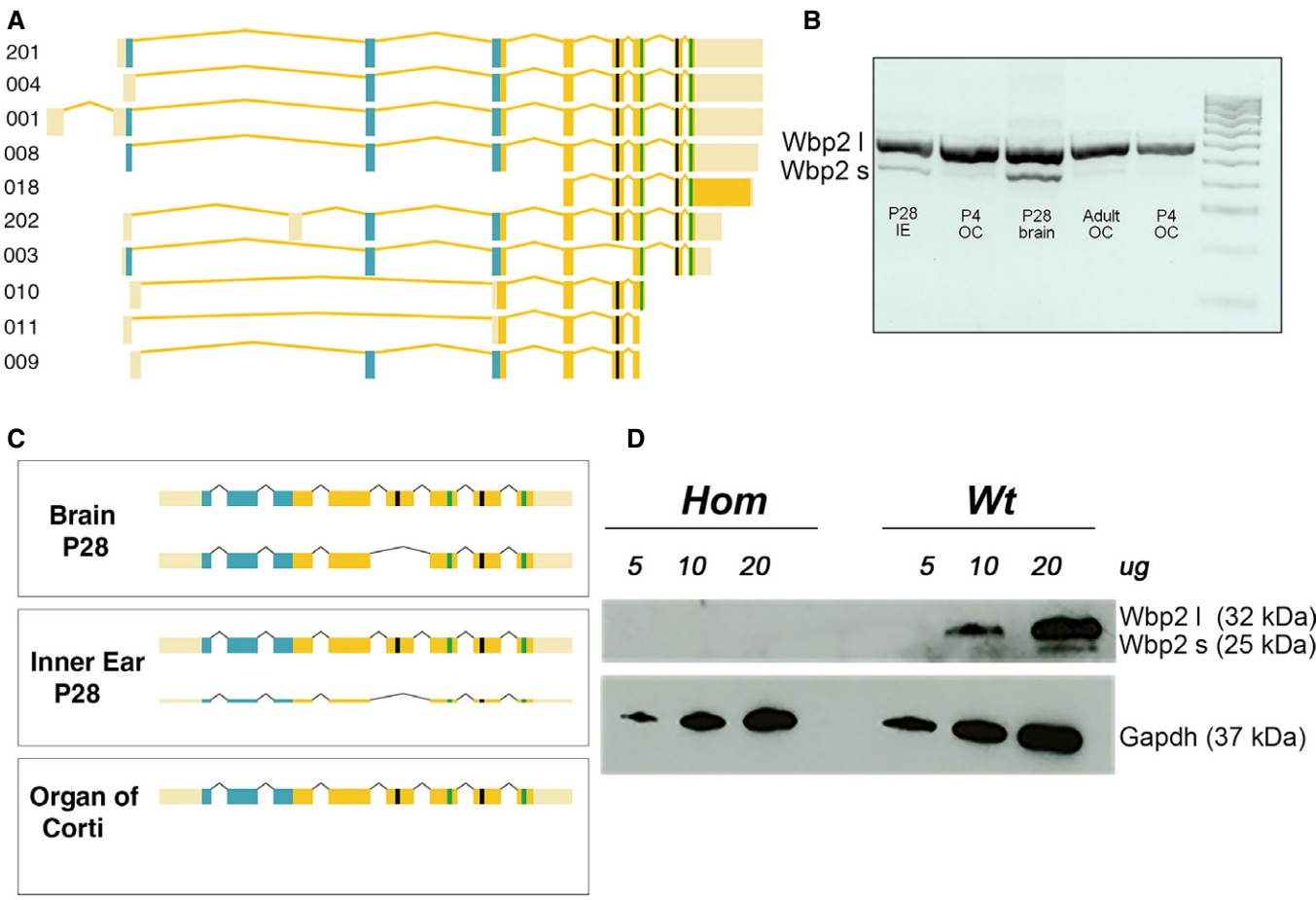

**Figure 4.  Analysis of the *Wbp2* mouse isoforms in the brain and in the cochlea.**

A  Splice forms of *WBP2*, numbered according to the Ensembl numbering scheme. The GRAM domain is marked in blue, the WW domains in green and the locations of the variants in black (the p.Ala160Thr and p.Met163Leu variants are too close to show separately in this view).

B  Agarose gel trace of cDNA obtained from mouse wt P28 inner ear (IE), P28 brain, adult and P4 organ of Corti (OC). Results show the expression of two *Wbp2* isoforms in the brain at P28 and very faint band for the short isoform together with a strong band for the long isoform in the inner ear at P28. If we look at just the organ of Corti (adult and P4), we observe a strong band for the long isoform in the P4 OC and an almost undetectable band for the short isoform, which was not even picked up by sequencing (see C). Wbp2 l: long isoform (550 bp); Wbp2 s: short isoform (480 bp).

C  Cartoon illustrating the results from the sequencing of mouse wt cDNA performed at P4 and P28. While in the brain we detect both *Wbp2* isoforms and in the organ of Corti only the long isoform, in the whole inner ear sample we detect the presence of the long isoform with a small band for the short one.

D  Western blot showing the predominant presence of the Wbp2 long isoform in the cochlea at P28, with a weak trace of the short isoform showing up only when a higher concentration of protein lysate (20 µg) is loaded. Both isoforms are absent in the Wbp2-deficient mouse. Gapdh was used as a loading control. Wbp2 l: long isoform; Wbp2 s: short isoform.

Source data are available online for this figure.

Liberman, 2009). We examined the complete length of the cochlea, but focused our quantification on the regions most sensitive to 6 and 24 kHz, corresponding to frequencies with normal and raised ABR thresholds, respectively.

Just after the onset of hearing (P14), we observed swollen afferent terminals on IHCs in the 24-kHz region of mutants compared to controls (Fig 5A). At 4 weeks, the changes were more evident (Fig 5A); in controls, the nerve terminals contacting IHCs are thin and filamentous and are aligned with ribbons, but in mutants, swelling of afferent terminals was observed under all IHCs and some unaligned ribbons were observed (Fig 5A). OHC afferent terminals were also swollen in the 24-kHz region at 4 weeks old (empty arrowheads in Fig 5A). The swelling of afferent terminals was also observed by transmission electron

microscopy (TEM) of the organ of Corti in mutants at P28 (Fig 5C). At 8 weeks, IHC innervation was even more severely affected in mutants (arrow in Fig 5D). The apical turn showed swollen terminals below IHCs at 4 and 8 weeks (Fig 5A and D); however, the swelling looks more pronounced in the 24-kHz region compared to 9-kHz region, fitting with the high-frequency hearing loss. Swelling of nerve terminals has been reported as a sign of glutamate excitotoxicity (Mayer & Westbrook, 1987) and described in various cochlear conditions such as ischaemia, neural presbycusis, noise-induced hearing loss and certain forms of sudden deafness or peripheral tinnitus (Puel *et al*, 1994, 1998; Kujawa & Liberman, 2009).

The number of ribbons per IHC in mutants was similar to that of controls at both frequency positions and at both 4 and 8 weeks of

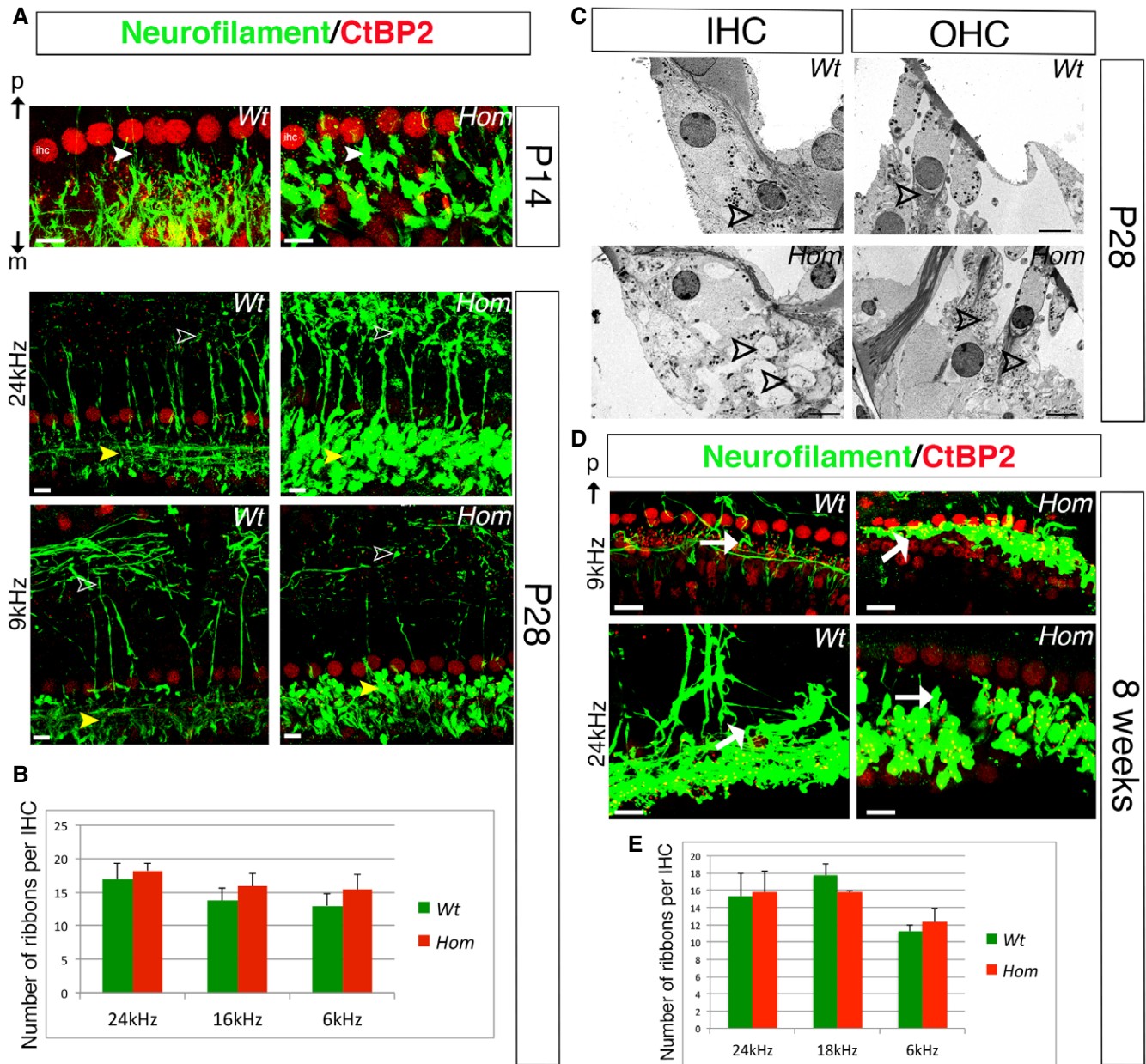

**Figure 5. Afferent innervation in Wbp2-deficient mice.**

A   At P14, afferent terminals below IHCs are slightly swollen in the mutants (neurofilament labelling in green, arrowheads to compare; CtBP2 labels ribbons and IHC nuclei in red). Scale bars, 10 μm. At P28, neurofilament/CtBP2 labelling in the organ of Corti of 4-week-old Wbp2-deficient mice and littermate controls shows severe swelling of IHC afferent terminals in the mutants, especially in the 24-kHz region (yellow arrowheads). The pre-synaptic ribbons do not look as well aligned to the terminals in the mutants (white arrows). At this stage, we also observe swelling of OHC afferent terminals in the 24-kHz region (empty arrowheads). Scale bars, 5 μm. ihc: IHC nucleus; p: pillar side; m: modiolar side.

B   Counts of pre-synaptic ribbons per IHC in the 8-, 18- and 24-kHz regions, showing no difference between mutants and controls at P28.

C   TEM of the organ of Corti performed at P28 showing swollen afferent terminals below inner and outer hair cells (arrowheads for comparisons between mutants and controls). Scale bars, 5 μm.

D   Neurofilament/CtBP2 labelling shows swollen and retracting terminals (white arrows), especially in the 24-kHz regions in the mutants at 8 weeks (arrows). Scale bars, 10 μm.

E   Counts of pre-synaptic ribbons in the 8-, 18- and 24-kHz regions, showing no difference in their number per IHC in the mutants compared to littermate controls at 8 weeks.

Data information: All data are shown as mean ± SD and statistically analysed by two-tailed Student's *t*-test. *n* = 35 hair cells. Synaptic count at 4 weeks: wild type: 24 kHz 16.9 ± 2.4, 16 kHz 13.67 ± 2, 9 kHz 13.0 ± 1.8; mutants: 24 kHz 18.17 ± 1.18 (*P* = 0.27), 16 kHz 15.89 ± 1.8 (*P* = 0.26), 9 kHz 15 ± 2 (*P* = 0.07). Synaptic count at 8 weeks: wild type: 24 kHz: 15.35 ± 3.75, 18 kHz 17.73 ± 1.79, 8 kHz: 11.25 ± 0.92; mutants: 24 kHz: 15.78 ± 3.46 (*P* = 0.9), 18 kHz 15.78 ± 0.17 (*P* = 0.26), 8 kHz: 12.3 ± 2.26 (*P* = 0.6)

Source data are available online for this figure.

age (Fig 5B and E), indicating that the ABR threshold shifts are not due to the loss of pre-synaptic ribbons.

At 4 weeks, GluR2/3 and CtBP2 double labelling showed reduced GluR2/3 expression and reduced overlap between ribbons and this post-synaptic density (PSD) marker in mutant IHCs compared to controls, suggesting a post-synaptic defect (Fig 6A). TEM analysis performed in the same cochlear region showed an array of different synaptic phenotypes in the mutant IHCs as shown in Fig 6B–G. We observed differentially shaped/sized ribbons, orphan PSDs and orphan ribbons, ribbons with attached synaptic membranes floating in the swollen afferent terminals, as well as close-to-normal looking synapses.

### Wbp2 controls the expression of the post-synaptic scaffolding proteins Psd-95 and Shank3 via transcriptional regulation of *Esr1, Esr2* and *Pgr*

We used our measured expression levels of key genes in the cochlea combined with published data to build a pathway to understand the mechanistic link between the loss of Wbp2 and progressive hearing loss (Fig 7A). Wbp2 acts as a transcriptional regulator of *Esr1* and *Pgr* (Dhananjayan *et al*, 2006; Lim *et al*, 2011). *Esr2* expression is linked with *Esr1* and *Pgr* expression (Lindberg *et al*, 2003; Aguirre *et al*, 2010). The estrogen receptor ERα (encoded by *Esr1*) binds to the *Shank3* promoter (Kwon *et al*, 2007), and signalling through the estrogen receptor ERβ (encoded by *Esr2*) increases the levels of Psd-95 and the AMPA receptor component GluR1 (Liu *et al*, 2008). Once activated by the hormone, estrogen receptors, which are expressed at PSDs (Adams & Morrison, 2003), are able to translocate into the nucleus and bind to DNA to regulate the activity of a number of post-synaptic genes (McEwen *et al*, 2001).

qRT–PCR showed a significant down-regulation of *Esr1, Esr2* and *Pgr* in 4-week-old *Wbp2*-mutant cochlea compared to littermate controls (Fig 7B), validating these regulatory links of Wbp2 *in vivo* in the auditory system. As predicted by the pathway analysis, we observed a decreased expression of GluR2/3 AMPA subunits in 4-week-old mutant cochlea (Figs 7A and 6A). We then tested the mRNA levels of *Shank3* and *Dlg4* (encoding Psd-95). Shank3 and Psd-95 are scaffolding proteins of AMPA and NMDA receptor subunits at the post-synaptic density of glutamatergic synapses, where they also play an important role in synaptogenesis and synaptic function (Migaud *et al*, 1998; Tu *et al*, 1999; Peca *et al*, 2011). Unexpectedly, we detected a significant increase in mRNA levels for both genes in the 4-week-old mutant cochlea (Fig 7B).

We assessed the level of Psd-95 protein by confocal imaging of sensory epithelia labelled with a Psd-95 antibody (Fig 7C). The up-regulation of Psd-95 mRNA in the organ of Corti of Wbp2-deficient mice (Fig 7B) was reflected in the protein level (Fig 7C and D).

## Discussion

We describe here for the first time a key role for Wbp2 in auditory function in humans and mouse and investigate the molecular mechanisms involved. Wbp2 is expressed widely in the cochlea with strong labelling of spiral ganglion neurons. Wbp2-deficient mice show progressive high-frequency hearing loss with little sign of hair cell degeneration and normal numbers of CtBP2-labelled ribbons.

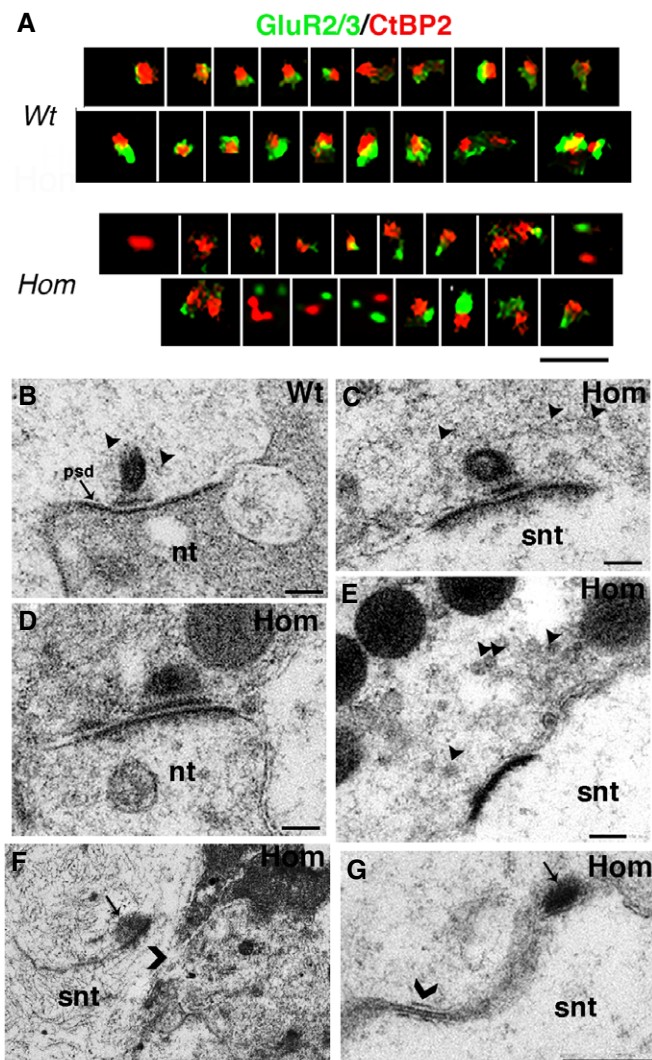

**Figure 6. GluR2/3 expression and synaptic defects in Wbp2-deficient mice.**

A    We arrayed synapse images aligned by ascending size of the post-synaptic site after GluR2/3 and CtBP2 labelling. This is a composite image made of several synapses taken from a single IHC from a single wt and a single hom, representing double labelling experiments performed on 3 mutants and 3 controls. Synapses in the mutants show abnormal morphology and smaller green patches, suggesting reduced expression of the GluR2/3 AMPA receptor subunits. Scale bar (shown on the bottom right), 1 μm.

B–G   TEM images of IHC ribbon synapses of wt (B, arrowheads for synaptic vesicles) and homs (C–G) at 4 weeks of age, showing a representative array of synaptic phenotypes in the 24-kHz cochlear region of mutants. While in (C) and (D) the ribbons look slightly abnormal in size with misplaced synaptic vesicles (arrowheads in C), we also observe orphan post-synaptic densities surrounded by floating synaptic vesicles (arrowheads in E); ribbons with synaptic membranes (arrow in F) that have detached from the IHC membrane (the arrowhead labels the original position of the synapse before detachment) and are floating in the swollen nerve terminal (F); ribbons (arrow in G) that are detached from their densities (arrowhead in G). Scale bars: (B–E), 200 nm; (F, G), 500 nm. nt: nerve terminal; psd: post-synaptic density; snt: swollen nerve terminal.

Source data are available online for this figure.

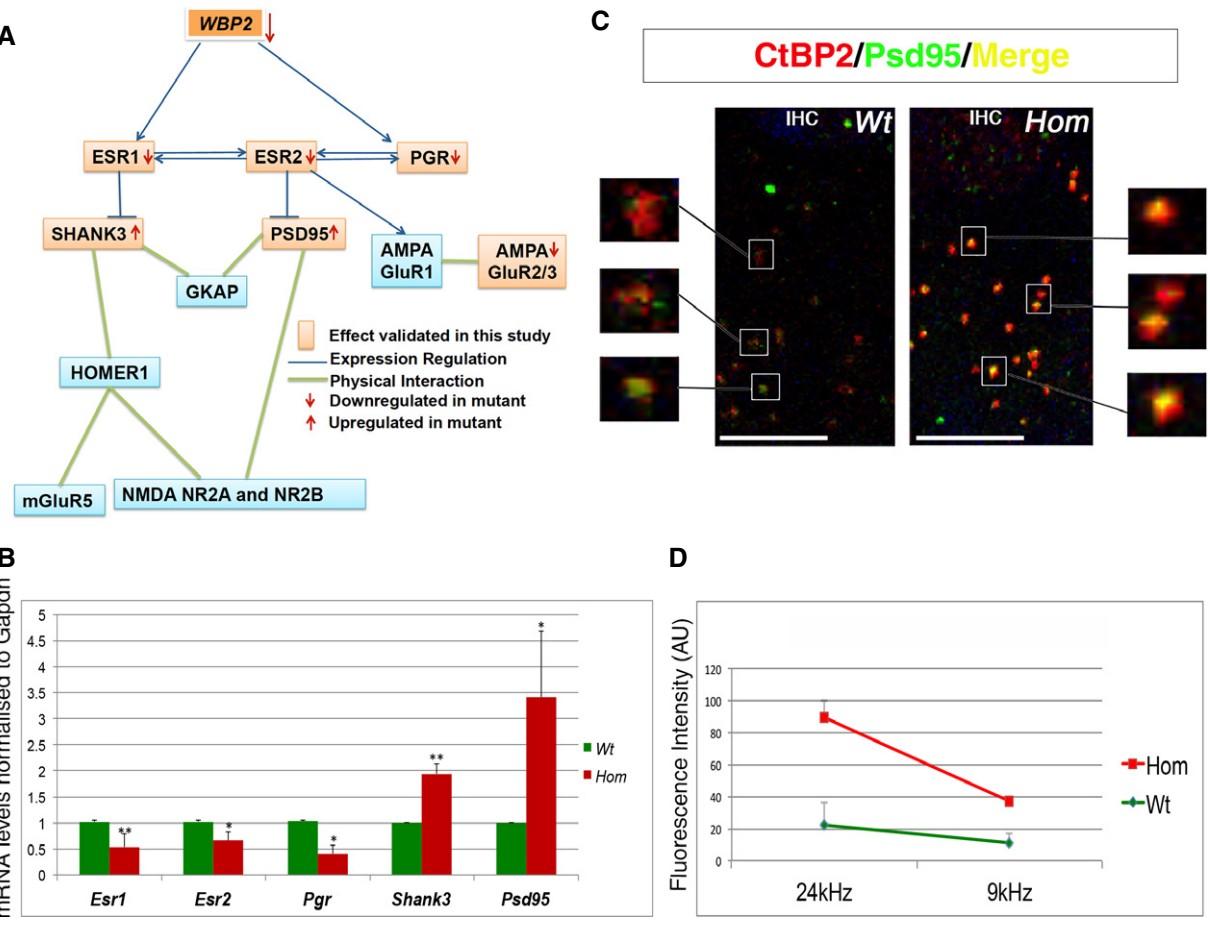

**Figure 7.  The Wbp2 molecular pathway.**

A   Diagram showing the Wbp2 molecular pathway, including its downstream targets and their functional relationship. The blue arrows, light blue lines and green lines link data from the literature (*in vivo* and *in vitro*); the orange squares and red arrows indicate up- or down-regulation shown in our experimental observations, as reported in this study.

B   Quantitative real-time PCR showing reduced mRNA levels for *Esr1*, *Esr2* and *Pgr* and up-regulation of *Shank3* and *Psd-95* in cochleae of 4-week-old Wbp2-deficient mice compared to littermate controls (*n* = 3 for each genotype). *Hprt* is used as a relative control. *$P$ = 0.03 for *Psd-95*; **$P$ = 0.007 for *Shank3*; *$P$ = 0.03 for *Esr2*; **$P$ = 0.0016 for *Esr1*; *$P$ = 0.037 for *Pgr*.

C   Synapses from one mutant and one control IHC at 4 weeks of age after Psd-95 and CtBP2 labelling, showing stronger Psd-95 expression in the mutants compared to controls, representing double labelling experiments performed on 3 mutants and 3 controls. Scale bar, 10 μm.

D   Quantification of Psd95 fluorescence in IHC synapses, representing expression in the apical (9-kHz best frequency region of the cochlea) and basal (24-kHz best frequency region of the cochlea) regions at 4 weeks of age. Data from 2 wt and 2 homs were analysed (16 synapses per cochlear region per mouse). AU, arbitrary units. Wt: 24 kHz 22.41 ± 10.70, 9 kHz 11.045 ± 2.128; mutants: 24 kHz 66.73 ± 13.70, $P$ = 0.069; 9 kHz: 26.02 ± 5.79 $P$ = 0.075.

Data information: Data are shown as mean ± SD and were statistically analysed by two-tailed Student's *t*-test.
Source data are available online for this figure.

Single IHCs show normal mechanoelectrical transducer current and capacitance changes upon depolarisation, suggesting normal neurotransmitter release. However, afferent nerve endings below IHCs are swollen and ribbon synapses show abnormal ultrastructural morphology associated with the disorganisation of the glutamate receptor complex at PSDs. PSDs labelled for GluR2/3 (Tu *et al*, 1999) appear reduced and show poor overlap with pre-synaptic ribbons, while the transcription of *Shank3* and *Dlg4* (encoding Psd-95) is increased. These changes can be explained by changes in oestrogen signalling, because *Esr1* and *Pgr* expression is reduced in the mutant cochlea and Wbp2 is known to act as a transcriptional coactivator of these genes. Estrogen and progesterone receptors are known to play a role in post-synaptic maintenance (Liu *et al*, 2008;

Baudry *et al*, 2013) and neuroprotection (Meltser *et al*, 2008; Simonoska *et al*, 2009).

**Wbp2-deficient mice exhibit progressive hearing loss due to a synaptic defect leading to cochlear excitotoxicity**

Hearing thresholds develop normally in *Wbp2*-mutant mice, but by 4 weeks of age, ABR thresholds are raised at high frequencies progressing with age to higher thresholds and lower frequencies. ABR waveforms of mutants showed reduced growth of wave 1 amplitudes with increasing stimulus level, compared with controls, suggesting a reduced auditory nerve activity. Our finding of reduced ABR amplitudes and abnormal IHC synapses in Wbp2-deficient mice

is similar to that reported following ageing or noise damage (Kujawa & Liberman, 2009; Sergeyenko et al, 2013). We also found raised thresholds for DPOAEs, indicating OHC dysfunction.

The most likely site of the initial IHC dysfunction is the post-synaptic region of the afferent synapse rather than the pre-synaptic complex, for a number of reasons. Firstly, resting membrane potential, basolateral membrane currents and neurotransmitter release from IHCs of Wbp2 mutants appear normal, and mutant hair cells show little sign of degeneration even at 30 weeks old. Secondly, GluR2/3 labelling, a marker of AMPA receptor subunits at PSDs, is reduced in mutant IHC synapses. Thirdly, the earliest sign of a structural defect is the swelling of the post-synaptic unmyelinated terminals, which progresses from P14 (when the synaptopathy is not severe enough to cause raised ABR thresholds) to 8 weeks of age, fitting with the progressive increase in ABR thresholds. Fourthly, the strongest intensity of Wbp2 expression is within spiral ganglion neurons, even though we do not see any obvious degeneration of these neurons in mice up to 6 weeks old. However, there are some signs of pre-synaptic defects too, including the reduced SP amplitude suggesting impaired depolarisation of mutant IHCs, and some misplaced ribbons observed by TEM, so we cannot rule out a direct effect of the mutation on IHCs. The raised thresholds for DPOAEs indicate abnormal OHC function, which will contribute to the raised ABR thresholds through reduced amplification by OHCs, but cannot directly explain the changes we detect at IHC synapses.

Interestingly, the disruption of the post-synaptic complex at IHC synapses in mutants does not lead to the loss of pre-synaptic ribbons, at least up to 8 weeks of age, but leads to swelling of IHC afferent nerve terminals, which is a sign of glutamate excitotoxicity. Glutamate can be toxic if it accumulates at excitatory synaptic clefts, giving rise to acute destruction of the nerve endings and eventually to neuronal death (Pujol & Puel, 1999). Swelling is a commonly described sign of synaptic impairment in the organ of Corti (Kujawa & Liberman, 2009).

## WBP2 is a novel human deafness gene

Wbp2 is haplosufficient in mice and in humans, as we found no abnormalities in mouse Wbp2 heterozygotes despite reduced mRNA levels (Fig 1C), and the four parents of the children with WBP2 variants had no reported hearing loss, suggesting that only one functional copy is required for normal hearing.

The results of our investigation into the isoforms present in the mouse organ of Corti and brain suggested that the long isoform, including exon 5, is the predominant isoform in the organ of Corti, while both long and short isoforms are present in the brain. It is therefore likely that variants affecting this long isoform, such as the p.Ala160Thr and p.Met163Leu variants, which both lie in exon 5, are more likely to impair critical protein function in the organ of Corti than in tissues that express both isoforms, such as the brain. This is in accordance with our observations of the children bearing compound variants in WBP2; each child carries the p.Ala160Thr variant, which is one of the two variants in exon 5. The first child carries a second variant in exon 7, which is predicted to be deleterious by most of the mutation predictors (Table 1). The second child has a second variant in exon 5, and although the predictions for the two exon 5 variants are less deleterious (Table 1), the importance of this exon for the organ of Corti supports our conclusion that the

deficiency in WBP2 is the cause of the deafness observed in these children.

## Steroid hormones and synaptopathy

Here, we defined the role of oestrogen signalling in the cochlea as essential for normal glutamatergic synapses in the cochlea. Both oestrogen and progesterone are known to be neuroprotective against stroke and glutamate toxicity in the central nervous system (CNS) when bound to their receptors (Singer et al, 1996; Liu et al, 2012). 17-β-Estradiol and its receptors have been reported to enhance synaptic function in the hippocampus, acting locally at PSDs, where Esr1 and Esr2 are also expressed (Adams & Morrison, 2003), controlling the expression of key post-synaptic proteins upon activation and translocation of the receptor into the nucleus (Akama & McEwen, 2003; Liu et al, 2008; Pinaud & Tremere, 2012). In the auditory system, oestrogen is known to control central and peripheral auditory processing (Hultcrantz et al, 2006). Esr1 and Esr2 are expressed in both IHCs and OHCs (Motohashi et al, 2010), and they protect against neuropathy following acoustic trauma (Meltser et al, 2008; Simonoska et al, 2009).

The levels of steroid hormones decrease during ageing (Motohashi et al, 2010); however, their levels can also be reduced in young individuals due to genetic or environmental causes, including Turner syndrome (Stenberg et al, 2002), autism (Sarachana et al, 2011; Rosenhall et al, 1999) and early-onset Alzheimer's disease (Henderson, 1997). In these cases, the main early symptoms are all associated with an impairment of synaptic plasticity and high-frequency hearing loss.

The down-regulation of Esr1, Esr2 and Pgr in Wbp2-deficient mice causes a significant decrease in GluR2/3 AMPA receptor subunits as well as a significant increase in the mRNA levels of the key post-synaptic proteins Shank3 (involved in autism and long-term potentiation) (Peca et al, 2011; Raynaud et al, 2013) and Psd-95 (with a key role in learning and memory) (Migaud et al, 1998). Their overexpression may be due to a negative feedback response to decreased estrogen receptor activation. In contrast, in non-pathological conditions, selective Esr2 agonists increase the expression of Psd-95 in the hippocampus (Liu et al, 2008).

The effects of Wbp2 deficiency are not limited to females. Wbp2-deficient mice are fertile, and we do not observe any differences in auditory function between genders. Moreover, deafness has been associated with variants in the human gene in both a boy and a girl. A brain-generated oestrogen binding to intracellular estrogen receptors at PSDs has been described because of its neurotrophic and neuroprotective actions in both males and females (Lee & McEwen, 2001; Motohashi et al, 2010; Pinaud & Tremere, 2012). Our finding of no significant abnormalities other than deafness in either mice or the children with Wbp2 deficiency suggests that Wbp2 might have a specific effect on the transcription of the intracellular estrogen receptors in the auditory system rather than on the hormonal receptors regulating gonadal functions. This opens up the Wbp2 pathway as a route to therapeutic approaches that specifically target the auditory system, avoiding potential unwanted effects of more general pharmacological manipulation of oestrogen/progesterone signalling.

We have demonstrated a clear link between the Wbp2 mutation and hearing loss. The lack of detectable Wbp2 results in reduced expression of Esr1, Esr2 and Pgr in the cochlea through the activity

of Wbp2 as a transcriptional coactivator of estrogen and progesterone receptors (Dhananjayan *et al*, 2006; Lim *et al*, 2011). Oestrogen signalling is known to control the expression of post-synaptic proteins upon activation and translocation of the receptor into the nucleus (Akama & McEwen, 2003; Liu *et al*, 2008; Pinaud & Tremere, 2012), and we found disrupted expression of Psd-95, Shank3 and AMPA receptor subunits in the *Wbp2*-mutant cochlea, including in IHC synapses. IHCs showed normal pre-synaptic function, but the earliest pathology detected was swelling of the post-synaptic nerve endings, a sign of glutamate excitotoxicity, indicating a primary post-synaptic defect. The progressive increase in ABR thresholds correlated both in time (age) and in location (point along the cochlear duct) with the swelling of IHC afferent nerve endings. Furthermore, the reduced amplitude of ABR wave 1 indicates a reduced cochlear nerve afferent activity. Swelling of cochlear afferent nerve endings directly affects single cochlear neuron function and summed auditory responses (Robertson, 1983), leading to hearing impairment.

The finding of a new molecule, Wbp2, involved in hearing highlights a potential new target for manipulation of the oestrogen signalling pathway specifically in the auditory system to prevent or reverse progressive hearing loss. These findings highlight the control of post-synaptic proteins as a key target of oestrogen signalling in the cochlea and suggest that manipulation of this pathway may be useful for treating other causes of synaptic damage due to other triggers, including later-onset progressive hearing loss. For example, the recent discovery that some types of age-related hearing loss as well as noise-induced damage can impair auditory function through primary auditory synaptopathy makes any insight into how this might be reversed particularly important (Kujawa & Liberman, 2009; Sergeyenko *et al*, 2013). Our findings also indicate which pathological subtype of hearing loss is most likely to respond to oestrogen-based therapy.

Finally, the finding of a new gene involved in human deafness after the original discovery of its role in hearing through a mouse screen emphasises the value of mouse genetics in understanding human disease.

# Materials and Methods

### Ethics statement

Mouse studies were carried out in accordance with UK Home Office regulations and the UK Animals (Scientific Procedures) Act of 1986 (ASPA) under UK Home Office licences, and the study was approved by the King's College London and the University of Sheffield Ethical Review Committees. Mice were culled using methods approved under these licences to minimise any possibility of suffering.

For human studies, informed consent was obtained from the adult participants and the parents or guardians of children prior to participation, and the experiments conformed to the principles set out in the WMA Declaration of Helsinki and the Department of Health and Human Services Belmont Report. The study was approved by the Ethics Committee of the First Affiliated Hospital of the Third Military Medical University, PLA, Chongqing, China.

### Statistics

Sample sizes were in line with power calculations carried out to detect biologically meaningful differences between mutants and controls for each measure using estimates of variance from previous similar experiments. Matched littermate controls were used without randomisation. Data collection was carried out blinded where feasible. Statistical tests were selected according to the nature of the data, e.g. normal or non-normal, repeated measures, etc. The SigmaPlot v12.5 was used to aid selection of the appropriate test. Estimates of variation are plotted on figures as standard deviations and were generally similar as a proportion of the mean in mutant compared with control groups. Experiments and analysis were carried out in compliance with the ARRIVE guidelines.

### Production and genotyping of *Wbp2*$^{tm2a/tm2a}$ mice

Wbp2-deficient (*Wbp2*$^{tm2a(EUCOMM)Wtsi}$) mice were produced at the Wellcome Trust Sanger Institute and carry a knockout-first conditional-ready allele (Skarnes *et al*, 2011; White *et al*, 2013) in which a promoterless cassette including *LacZ* and *neo* genes was inserted in intron 1-2 of the *Wbp2* gene located on chromosome 11 (Fig 1A). The mice were maintained in individually ventilated cages at a standard temperature and humidity and in specific pathogen-free conditions on the mixed C57BL/6N genetic background, and the mice of both genders were used. The Wbp2-deficient mice are viable and fertile, and the frequency of homozygous offspring from heterozygous intercross matings follows the expected Mendelian ratio. Homozygous mutants were screened by ABR for hearing impairment at 14 weeks of age as part of a standardised battery of primary phenotypic tests (White *et al*, 2013).

To genotype animals, DNA was extracted from the tissue of ear-clips and used as the template for short-range PCR using the forward primers: for the wild-type allele: forward: GCCCAATGGAGAGGAA-CAAG and reverse: GTAACTCCAGCATCAGGGGG.

The mutant allele shares the same reverse primer with wild type (the cassette is inverted in the *Wbp2*$^{tm2a/tm2a}$ mice), and the reverse primer: TCGTGGTATCGTTATGCGCC.

The mutant line is available from the European Mutant Mouse Archive, EMMA (http://strains.emmanet.org/).

### Auditory brainstem response (ABR) recordings

ABRs were recorded in mice of a range of ages from post-natal day (P) 14 to 44 weeks old. Mice were anaesthetised with ketamine (1 mg/g)/xylazine (0.01 mg/g), and ABRs were recorded as previously described (Ingham *et al*, 2011). Response thresholds for each stimulus were estimated from the resulting ABR waveform and defined as the lowest sound level where any recognisable feature of the waveform was visible. Wave 1 amplitude and latency were measured with the help of ABR Notebook software routines (courtesy of MC Liberman, Harvard/MIT). In addition, we estimated amplitudes of the summating potential (SP, defined as a positive deflection peak occurring before wave 1). Due to the small amplitude of the SP, we generated an averaged waveform across all wt controls and all Wbp2-deficient mice recorded, using 5-dB SL increments.

There was no effect of gender on the range of 24-kHz thresholds recorded at 4 weeks old in male ($n = 19$, median = 30 dB)

or female wt ($n = 20$, median = 30 dB) (Mann–Whitney *U*-Statistic = 174.0, $T = 396.0$, $P = 0.652$) or in male ($n = 16$, median = 40 dB) or female hom ($n = 12$, median = 50 dB) (Mann–Whitney *U*-Statistic = 166.5, $T = 302.5$, $P = 0.975$). Similarly, there was no effect of gender on the range of 24-kHz thresholds recorded at 14 weeks old in male ($n = 2$, median = 20 dB) or female wt ($n = 8$, median = 25 dB) (Mann–Whitney *U*-Statistic = 2.0, $T = 5.0$, $P = 0.178$) or in male ($n = 8$, mean = 53.1 dB) or female hom ($n = 6$, mean = 77.5 dB) (*t*-test, $t = 2.065$, two-tailed $P = 0.0613$).

To exclude the possibility that the ketamine/xylazine anaesthetic combination may have selectively influenced the hearing phenotype of the post-synaptically compromised *Wbp2* mutants by blocking the activation of NMDA receptors at PSDs, we tested a different cohort of mice at P27–28 under urethane anaesthesia. ABR thresholds were estimated under urethane anaesthesia, given as 0.1 ml/10 g of a 20% solution, by i.p. injection. ABR waveform shapes were collated at 20 dB and 50 dB above threshold, and the mean waveform amplitude was calculated and plotted. Mean ABR thresholds in *Wbp2* homs were comparable to those seen in mutants tested under ketamine/xylazine, and the range of ABR thresholds of affected frequencies (24–42 kHz) was not significantly different under urethane anaesthesia ($n = 9$) compared to ketamine/xylazine ($n = 37$) (Mann–Whitney *U*-Statistic = 2,374.0, $P = 0.303$) (Fig 2B).

### Distortion product otoacoustic emission (DPOAE) measurements

We measured distortion product otoacoustic emissions in mice aged 4 and 21 weeks old, anaesthetised with 20% urethane. Experiments were performed using Tucker Davis Technologies (TDT) BioSigRZ software driving a TDT RZ6 auditory processor and a pair of TDT EC1 electrostatic transducers. Signals were recorded via an Etymotic ER-10B+ low-noise DPOAE microphone. Stimuli were presented and microphone signals recorded via a closed-field acoustic system sealed into the auditory meatus of the mouse. Stimulus tones were presented in an f2:f1 ratio of 1.2. f2 tones were presented at frequencies matching ABR measurements (6, 12, 18, 24 and 30 kHz). f1 was presented at levels from 0–75 dB in 5-dB steps. f2 was presented at 10 dB below the level of f1. The magnitude of the 2f1-f2 DPOAE component was extracted from a fast Fourier transform of the recorded microphone signal and plotted as a function of f2 level. For each f2 level, the 20 spectral line magnitudes surrounding the 2f1-f2 frequency were averaged to form a noise floor for each measurement. DPOAE threshold was defined as the lowest stimulus level where the emission magnitude exceeded 2 standard deviations above the mean noise floor.

### Endocochlear potential (EP)

Mice were anaesthetised with an i.p. injection of urethane (0.1 ml/10 g of a 20% solution), a tracheal cannula was inserted and the mouse placed in a head holder on a homeothermic heating blanket. EP was measured as previously described (Steel & Barkway, 1989; Chen *et al*, 2014).

### RNA extraction and quantitative RT–PCR

The cochleae of 4-week-old *Wbp2*-mutant mice and littermate controls ($n = 3$ for each genotype) were dissected and stored at

−20°C in RNAlater stabilisation reagent (QIAgen, cat. no. 76106). RNA was extracted using QIAshredder columns (QIAgen, cat. no. 79654) and the RNeasy mini kit (QIAgen, cat. no. 74104), following the manufacturer's instructions. cDNA was created using Superscript II Reverse Transcriptase (Invitrogen, cat. no. 11904-018) after treatment with DNase 1 (Sigma, cat.no: AMP-D1). Real-time PCR was performed in an CFX Connect Real-Time System (Bio-Rad), in triplicate for each sample using the following TaqMan probes from Applied Biosystems: Mm01246338_m1 (*Wbp2*); Mm00498775_m1 (*Shank3*); Mm00492193_m1 (*Psd-95*); Mm00433149_m1 (*Esr1*); Mm00599821_m1 (*Esr2*); Mm00435628_m1 (*Pgr*). The PCR was performed. Hypoxanthine-guanine phosphoribosyltransferase (*Hprt*) was amplified simultaneously (Applied Biosystems, Mm01318747_g1) as the internal reference. Statistical analysis was performed using the Student's *t*-test.

### SNP prediction

Fathmm (http://fathmm.biocompute.org.uk/, (Shihab *et al*, 2013)), Mutation Assessor (http://mutationassessor.org/, (Reva *et al*, 2011)), Mutation Taster (http://www.mutationtaster.org/, (Schwarz *et al*, 2014)), SIFT (http://sift.jcvi.org/, (Ng & Henikoff, 2003)), PolyPhen (http://genetics.bwh.harvard.edu/pph2/index.shtml, (Adzhubei *et al*, 2010)) and PhD-SNP (http://snps.biofold.org/phd-snp/phd-snp.html, (Capriotti *et al*, 2006)) were used to predict the effects of the human variants upon WBP2 protein function. In all cases, the long isoform was used (ENST00000254806; ENSP00000254806). SIFT offers multiple methods of analysis, so we used the implementation in Ensembl's Variant Effect Predictor (http://www.ensembl.org/Homo_sapiens/Tools/VEP?db = core), the automated method (http://sift.jcvi.org/www/SIFT_enst_submit.html) and the manual method where we selected the input orthologues (http://sift.jcvi.org/www/SIFT_related_seqs_submit.html).

### cDNA sequencing

cDNA was obtained from the organ of Corti of P4 wild-type mice ($n = 3$), adult organ of Corti ($n = 1$) and from cochleae of P28 wild-type mice ($n = 4$) as described above. Brains were extracted from P28 wild-type mice ($n = 4$), snap-frozen in liquid nitrogen and stored at −20°C. RNA was extracted using TRIzol, and cDNA was made as described above. Primers were designed for sequencing using Primer3 (Untergasser *et al*, 2012), (Koressaar & Remm, 2007)): ENSMUSE00000252777_F: CTGTGAGATTAAGCAGCCGG, ENSMUSE00000669944_R: GTAAGAGGCGGGAAGTGGG. Capillary sequencing was carried out by Source BioScience (Nottingham, UK), and the traces were examined using Gap4 (Bonfield *et al*, 1995).

### Western blot

Protein lysates from half-brain and from six cochleae pooled together across three different mice with the same genotype ($n = 3$ of each genotype) at 4 weeks of age were analysed by Western blot on 10% SDS–PAGE (Buniello *et al*, 2013). The primary antibodies used were: Wbp2 (Cell Signaling, 11831; 1:1,000), GluR2/3 (Millipore, 7598; 1:1,000), Gapdh (Abcam, ab9482; 1:5,000) and β-tubulin (Covance, MRB-435P; 1:5,000). The secondary antibodies used were anti-rabbit HRP conjugate (Millipore, 12-348; 1:5,000) and

anti-mouse HRP conjugate (Bio-Rad, 170-6516; 1:5,000). While the cochlear Westerns were developed using a conventional darkroom, the brain Westerns were developed using a Biospectrum 810 imaging system (UVP, Upland, CA).

### Semi-thin sections and transmission electron microscopy

Inner ears ($n = 3$ for each genotype) were dissected and processed as previously described (Chen *et al*, 2014). Ultrathin sections (70–90 nm) were cut longitudinally using a Leica EM UC7 ultramicrotome with knife angle of 35 degrees, mounted on copper grids and contrasted using uranyl acetate and lead citrate. Samples were examined on a JEOL JEM-1400Plus (120 kV) fitted with an 8-M pixel fully integrated high-precision camera. Semi-thin sections (0.5 μM) were stained with 1% toluidine blue for light microscopy observation.

### Histology and anatomy of the ear

Bisected heads from mutants and littermate controls aged 4–6 weeks ($n = 3$ for each genotype) were fixed with 10% formalin and embedded in paraffin wax, and 8-μm sagittal sections were stained with haematoxylin and eosin. The anatomy of the middle ear was examined as previously described (Chen *et al*, 2013), and the specimens were imaged using microscope with Nomarski optics (Axioplan; Zeiss) and digital camera (AxioCam; Zeiss).

### X-gal staining

In the knockout-first conditional-ready design used to generate the *Wbp2*-mutant allele (Fig 1A), a *LacZ* gene inserted in the allele makes it possible to visualise the expression of the targeted gene by using X-gal. Wild-type littermates were used as negative controls. Inner ears from homozygous and heterozygous mutants and wild-type littermate controls ($n = 2$ for each genotype) at post-natal day 14 were dissected out and fixed in fresh 4% paraformaldehyde for 45 min at 4°C with rotation, washed in PBS and decalcified overnight in 10% EDTA before a detergent wash (2 mM $MgCl_2$; 0.02% NP-40; 0.01% sodium deoxycholate; in 0.1 M sodium phosphate buffer, pH 7.3 in PBS) for 30 min at room temperature.

X-gal (Promega; cat.no. V394A) was added 1:50 to 500 μl of pre-warmed staining solution (5 mM $K_3Fe(CN)_6$; ferrate (III) and 5 mM $K_4Fe(CN)_6$; ferrate (II)); then, ears were stained at 37°C in the dark for 1 h, washed in PBS and gradually dehydrated before paraffin embedding; 8-μm-thick sections were obtained from the embedded samples, which were counterstained with Nuclear Fast Red as the X-gal reaction labelling is always cytoplasmic (VWR, cat.no. 342094W), mounted on slides and imaged using the AxioCam MRc camera (Carl Zeiss), using the Axiovision 3.0 software (Carl Zeiss).

### Scanning electron microscopy

The organ of Corti from wild-type and mutant mice at 5 weeks (wt controls, $n = 11$; het, $n = 8$ and mutants $n = 7$) and 30 weeks (wt controls, $n = 15$; mutants, $n = 14$) of age was fixed and processed by the OTOTO protocol as previously described (Hunter-Duvar, 1978) and viewed with a Hitachi FE S-4800 Scanning Electron Microscope operated at 3–5 kV. The best frequency regions were determined as previously reported (Müller *et al*, 2010).

### Confocal imaging and post-acquisition analysis

For whole-mount immunostaining, cochleae at P14 ($n = 3$ for each genotype), 4 weeks ($n = 9$ for each genotype for neurofilament/CtBP2, $n = 3$ for GluR2/3/CtBP2 and $n = 3$ for Psd95/CtBP2) and 8 weeks ($n = 3$ for each genotype) were perfused with 4% paraformaldehyde, post-fixed for 1–2 h at room temperature, decalcified in EDTA for 24–48 h at 4°C and fine-dissected. Samples were then blocked in 5% normal horse serum (NHS), 1% bovine serum albumin (BSA) and 0.3% Triton X-100 in PBS for 1 h at room temperature and immunostained in 1% normal horse serum (NHS), 0.3% Triton X-100 in PBS as previously reported (Kujawa & Liberman, 2009). Antibodies against CtBP2 (BD Transduction Laboratories, 612044, 1:200), GluR2/3 (Millipore, 7598; 1:300), NF-H (Abcam, ab4680, 1:800), Myo7a (Proteus, 1:300) and Psd-95 (Cell Signaling Technology, 2507; 1:500) were used. Secondary antibodies used were Alexa Fluor 546 donkey anti-mouse (Invitrogen, A100036, 1:400), Alexa Fluor 488 goat anti-rabbit (Invitrogen, A11008, 1:400) and Alexa Fluor 488 goat anti-chicken (Invitrogen, A100036, 1:400). Mounting was performed using ProLong Gold mounting medium (Invitrogen, P36930) on 1-mm-thick microscope slides (Thermo Fisher Scientific, 1014356290F). Most confocal imaging was performed using a Nikon A1R point-scanning confocal microscope system (Nikon Instruments UK). The CtBP2 and GluR2/3 double immunofluorescence was imaged using a Leica TCS SP5 confocal microscope (Leica, Germany). All images were captured using Plan Apo VC 60x/1.4NA and Plan Apo VC 100x/1.4NA lenses.

Confocal *z*-stacks were obtained with a *z*-step size of 0.200 to 0.500 mm. NIS Elements v4.2 software (Nikon Instruments UK) and Image J (NIH, Bethesda, Maryland, US) software were used for acquisition and post-acquisition analysis, including synaptic counts.

The PSD-95 fluorescence intensity units at IHC synapses were quantified using the "Automated Measurements" tool of the NIS Elements software ($n = 2$ of each genotype). The reads were then averaged, and the statistical analysis was performed using the Student's *t*-test. The study was performed throughout the whole length of the cochlea, and representative images from the regions of interest are shown in the figures. The best frequency regions were determined as previously reported (Müller *et al*, 2010).

### Pathway analysis

The Wbp2 pathway analysis was generated with the help of QIAGEN's Ingenuity Pathway Analysis (IPA®, QIAGEN Redwood City, www.qiagen.com/ingenuity). All links were checked by reading the relevant primary publications before including in the pathway. Additional data from our own experiments were added.

### Single-hair cell electrophysiology

Inner hair cells from control and littermate Wbp2-deficient mice were studied in acutely dissected organs of Corti from post-natal day 7 (P7) to P33, where the day of birth is P0. The basal coil of the cochleae, corresponding to an approximate frequency range of 35–55 kHz, was dissected in normal extracellular solution (in mM):

**The paper explained**

**Problem**

Progressive hearing loss is a very common disease. However, very little is known about its molecular mechanisms and the lack of knowledge leads to the lack of targets for medical therapies. Steroid hormones are known to be implicated in normal auditory function and neuro-protection, and estrogen receptors are expressed in sensory hair cells.

**Results**

In order to investigate the functional link between hormonal signalling and hearing impairment and identify new targets for therapies, we analysed Wbp2-deficient mice. *WBP2* encodes the WW domain-binding protein 2, which acts as a transcriptional coactivator for the estrogen and progesterone receptors *ESR1* and *PGR*.

We demonstrate for the first time a direct link between the *Wbp2* mutation and progressive hearing loss. Wbp2-deficient mice show progressive high-frequency hearing loss due to a primary synaptopathy. The lack of detectable Wbp2 results in reduced expression of *Esr1*, *Esr2* and *Pgr* in the cochlea. Oestrogen signalling is known to control the expression of post-synaptic proteins upon activation and translocation of the receptor into the nucleus, and we found disrupted expression of Psd-95, Shank3 and AMPA receptor subunits in the *Wbp2*-mutant cochlea, including in inner hair cell synapses. The dissociation of the post-synaptic complex explains the synaptopathy and therefore the hearing impairment. The earliest pathology detected was swelling of the post-synaptic afferent nerve endings at inner and outer hair cells, a sign of glutamate excitotoxicity. Glutamate can be toxic if it accumulates at synaptic clefts as a result of a synaptic dysfunction. The progressive increase in the hearing impairment correlated both in time (age) and in location (point along the cochlear duct) with the swelling of IHC afferent nerve endings.

Most importantly, we also demonstrated that *WBP2* is crucial for hearing also in humans. We reported the cases of two children affected by severe to profound deafness, each carrying two variants in heterozygosis in the *WBP2* gene.

**Impact**

This study opens up the Wbp2 pathway as a route to therapeutic approaches that specifically target the auditory system, avoiding potential unwanted effects of more general pharmacological manipulation of oestrogen/progesterone signalling.

135 NaCl, 5.8 KCl, 1.3 $CaCl_2$, 0.9 $MgCl_2$, 0.7 $NaH_2PO_4$, 5.6 D-glucose, 10 Hepes-NaOH. Sodium pyruvate (2 mM), MEM amino acids and vitamin solutions were added from concentrates (pH 7.5; ~308 mmol/kg).

Electrophysiological recordings were performed using an Optopatch (Cairn Research Ltd, UK) or Axopatch 200B (Molecular Devices, USA) amplifiers. Data acquisition was controlled by pClamp software using Digidata 1440A boards (Molecular Devices, USA). Patch pipettes were coated with surf wax (Mr Zoggs SexWax, USA) to minimise the fast patch pipette capacitance transient. Recordings were performed either at room temperature (22–24°C) or at body temperature (34–37°C).

Current and voltage recordings from IHCs (see Table EV1) were performed using the following intracellular solution in the patch pipette containing (in mM): 131 KCl, 3 $MgCl_2$, 1 EGTA-KOH, 5 $Na_2ATP$, 5 Hepes-KOH, 10 $Na_2$-phosphocreatine (pH 7.3; ~296 mmol/kg). Recordings were low-pass-filtered at 2.5 kHz (8-pole Bessel), sampled at 5 kHz and stored on computer for off-line

analysis (Origin: OriginLab, USA). Membrane potentials in voltage clamp were corrected for the voltage drop across the uncompensated residual series resistance ($R_s$: 1.0 ± 0.1 MΩ, $n$ = 15) and for a liquid junction potential (LJP) of −4 mV.

Mechanoelectrical transducer (MET) currents were elicited by stimulating the hair bundles of OHCs using a fluid jet from a pipette (tip diameter of 8–10 μm) driven by a piezoelectric disc (Corns *et al*, 2014). V. MET currents were recorded with a patch pipette solution containing (in mM): 106 Cs-glutamate, 20 CsCl, 3 $MgCl_2$, 1 EGTA-CsOH, 5 $Na_2ATP$, 0.3 $Na_2GTP$, 5 Hepes-CsOH, 10 sodium phosphocreatine (pH 7.3). Membrane potentials were corrected for the LJP of −11 mV.

Real-time changes in membrane capacitance ($\Delta C_m$) were measured using the Optopatch as previously described (Johnson *et al*, 2008, 2010). Briefly, a 4-kHz sine wave of 13 mV RMS was applied to IHCs from −81 mV and was interrupted for the duration of the voltage step (50 ms and in 10 mV increments). The capacitance signal from the Optopatch was amplified (×50), filtered at 250 Hz and sampled at 5 kHz. $\Delta C_m$ was measured by averaging the $C_m$ trace over a 200-ms period following the voltage step. These exocytotic recordings were performed using the above Cs-based intracellular solution. Membrane potentials were corrected for the uncompensated residual series resistance ($R_s$: 6.0 ± 0.2 MΩ, $n$ = 25) and LJP (−11 mV). $\Delta C_m$ was recorded while applying 30 mM TEA and 15 mM 4AP (Fluka, UK) and additionally 80 μM linopirdine and 300 nM apamin (Tocris, UK) to reduce $K^+$ currents.

**Variant identification in the *WBP2* gene in human deafness**

We designed a target enrichment (TGE) kit for the molecular diagnosis of hereditary hearing loss in humans. The target region included a total of 438 human protein-coding genes potentially related to hearing loss. We adopted Agilent SureSelect TGE technology to manufacture the assay chemistries. The TGE kit was used to capture the genomic DNA from the probands with hearing loss, according to the supplier's instructions. The library enriched by the TGE kit was sequenced on an Illumina HiSeq 2000 in 90-bp paired-end reads. Raw image files were processed by Illumina CASAVA Software version 1.7 for base calling with default parameters. The bioinformatics analysis was carried out as described previously (Lu *et al*, 2014). Following the large-scale sequencing, specific primers were designed to amplify the genomic region encompassing exons 5 and 7 of the *WBP2* gene for the probands and the parents. PCR products were sequenced in both forward and reverse directions on an ABI 3100 using the BigDye Terminator Cycle Sequencing Ready Reaction Kit (Applied Biosystems).

**Expanded View** for this article is available online.

**Acknowledgements**

We thank Jing Chen for help with the human variant study, Selina Pearson for the initial ABR screen, John Harris for advice on post-acquisition image analysis and Johanna Pass, Zahra Hance, Lorenzo Preite and Victoria Rook for laboratory support. We thank M C Liberman for generously providing software to assist waveform analysis. We are grateful to Rosalind Lacey and Claire Pearce for the animal care and to the Wellcome Trust Sanger Institute Mouse Genetics Project team for the production of the *Wbp2*-mutant mice. This work was supported by the Wellcome Trust to KPS (098051 and 100669) and WM

(102892) and by the National Science Fund for Distinguished Young Scholars (81125008) and the National Basic Research Program (2013CB945402) to HY.

## Author contributions

AB codesigned the research; performed expression analysis, confocal studies and post-acquisition analysis, Western blots, histological studies, TEM, qRT–PCR, building of the molecular pathway, and analysed the data. NJI codesigned the experiments and performed ABR, DPOAE and EP recordings and analysis. ACH contributed to the cochlear innervation experiments and qRT–PCR. RM-V performed ABR, SEM and anatomical characterisation of middle and inner ear. IV-N supported the work of RM-V. MAL worked on the *in silico* characterisation of the human variants and isoform analysis. OH, TB, SLJ and WM performed the single-cell electrophysiological experiments and analysis. GV-B carried out the TEM. RAF supported the work of GV-B. HY carried out the targeted sequencing screen of deaf individuals, identified the human variants and provided the clinical data. JKW contributed to the production of the *Wbp2* mutant and managed the Mouse Genetics Project. KPS conceived the study, supervised the experiments, the analysis and the interpretation of the data. KPS, AB and WM wrote the paper. All authors discussed the results and commented on the manuscript.

## Conflict of interest

The authors declare that they have no conflict of interest.

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
