## [Review Process File · EMBO Molecular Medicine]

Wbp2 is required for normal glutamatergic synapses in the cochlea and is crucial for hearing

Annalisa Buniello, Neil J Ingham, Morag A Lewis, Andreea C Huma, Raquel Martinez-Vega, Isabel Varela-Nieto, Gema Vizcay-Barrena, Roland A Fleck, Oliver Houston, Tanaya Bardhan, Stuart L Johnson, Jacqueline K White, Huijun Yuan, Walter Marcotti, Karen P Steel

Corresponding author: Annalisa Buniello and Karen P Steel, Kings College London

Review timeline:

Submission date:	21 April 2015
Editorial Decision:	05 June 2015
Transfer Date:	09 June 2015
Editorial Decision:	11 June 2015
Revision received:	11 November 2015
Additional Author Correspondence	13 December 2015
Editorial Decision:	14 December 2015
Revision received:	18 December 2015
Accepted:	21 December 2015

Transaction Report:

Editor: Céline Carret

1st Editorial Decision

05 June 2015

Thank you for submitting your manuscript to the EMBO Journal. Your study has now been seen by two referees and I am afraid that their overall recommendation is not very positive.

While the referees appreciate the finding that loss of Wbp2 leads to hearing loss in mice and humans, they also raise issues with the analysis that I am afraid preclude publication here. They find that both the clinical aspects of the work and the mechanistic insight into how loss of Wbp2 leads to hearing loss is not sufficiently developed in order to consider publication here. Given these comments from good experts in the field, I am afraid that I can't offer to consider publication here.

Given the clinical relevance of the findings, I have also discussed the manuscript further with my colleague Celine Carret (c.carret@embomolmed.org) at EMBO Molecular Medicine (<http://embomolmed.embopress.org/>). EMM is interested in considering the manuscript for publication and can work with the two existing referee reports. For EMM, the clinical issues would have to be sorted out, but not the mechanistic ones. If you are interested in this option please contact Celine directly to discuss this further.

For the EMBO Journal. I am afraid that I can't be more positive on this occasion, but I do hope that you will consider the EMM option.

REFeree REPORTS

Referee #1:

The study by Buniello et al reports that loss of Wbp2 function in mice results in hearing loss and the identification of a child with compound heterozygosity in the Wbp2 gene that presents with profound deafness. Furthermore, using physiological histological analysis, they go on to show that the inner ear of Wbp2 mutant mice shows a progressive disruption of the postsynaptic terminals innervating inner hair cells. Moreover, they document that KO of Wbp2 results in dysregulation of estrogen related and post-synaptic genes. Based on these observations, the authors conclude that Wbp2 is a critical player in hearing and propose that the synaptic defects result from excitotoxic events downstream of the increased expression of post-synaptic molecules such as PSD95 and Shank3.

The experiments are well done and the data provides a strong link between Wbp2 and progressive hearing loss in mice. However, many of the claims and conclusions are not supported by the data. Furthermore, the relevance of Wbp2 mutations for hearing loss in humans is tenuous.

Major comments:

1. The paper does not really explain how the KO mice were generated. Specifically, the paper provides a depiction of the construct (Fig 1A), but I could not find any place where they say if exon 2 was knocked out by Cre-mediated recombination and how this have been done, neither if they phenotypic analysis was done in mice carrying the LacZ/neo cassette or if those sequences were flipped out.
2. It is surprising that the authors did not test outer hair cell function by DPOAEs in these mice. This is essential to understand the hearing phenotype.
3. While the evidence that Wbp2 LOF results in progressive hearing loss in mice is compelling, the mechanistic hypothesis about how this occurs are based purely in correlations that are intriguing but not solid or defensible.
4. The authors claim that the phenotype in the Wbp2 mutant mice resembles cochlear excitotoxicity observed after noise exposure or ischemia, with common pathological hallmark of cochlear nerve terminal swelling. However, the authors observed a normal juxtaposition between *ctbp2* and *psd-95* (Fig7C), suggesting that the nerve terminal is still in contact with the presynaptic terminals. This is an intriguing observation considering the massive terminal swelling in the mutants. To understand the pathology it will be necessary to interrogate the structural features of the synaptic terminals using EM.
5. The data on the summing potential (SP) (Fig 2I) has very large error bars, making it hard to understand how it is significantly different. Similarly, the images used to claim alterations in inner hair cell ribbon distribution are not convincing. 3D reconstructions might help make the point more clear and compelling.
7. In animal models, complete *wbp2* knockout results in progressive hearing loss that is mild at young. However, in the 5-yr human subject, the two *wbp2* alleles associate with profound childhood deafness. It is not sufficiently convincing that the two *wbp2* mutations could explain the disease phenotype. The authors should explain this concern. In addition, unlike what the authors described: "The A224V mutation is novel and predicted to affect protein function (SIFT; score=0.00)", SIFT analysis with ENSP number (ENSP00000254806) predicts A224V to be "TOLERATED" for protein function, with SIFT score of 0.3.
8. It would be important to test if the interaction between Wbp2 LOF and changes in expression of the other genes (*Psd-95*, *Shank3*, *Esr1*, *Esr2* and *Pgr*) are specific to the inner ear or are relevant to other parts of the nervous system, and determine the reason for the specificity if it exists. If not, it would be important to explain why other excitatory synapses are not affected.

Referee #2:

General Comments:

This manuscript describes peripheral auditory changes in *Wbp2*-deficient mice. The authors have performed numerous assays to characterize the peripheral auditory organ in these mice. While the careful characterization of this mouse mutant is commendable, it tends to distract from the main point of the study. Many of these assays showed no differences between the mutants and the wild-type mice and these findings should be placed in the supplementary section. The section with the genetic mutation in the 5 year old boy is interesting, but without any audiometric tests or other types of characterization this data seems to hang in the air and should be removed from the manuscript.

The sections of the manuscript where there are differences between the two groups are not well performed or presented. It is not understood why the authors have selected two narrow regions to study the post-synaptic changes (i.e. the main part of the study). The authors concentrate on 9 and 24 kHz that corresponds to the apex and towards the base of the cochlea. Quantification should have been performed throughout the entire cochlea for the data showing differences between the two groups. How can the authors be sure that there are no changes in the other regions of the cochlea? This is a weak aspect to the study.

Specific Comments:

The ABR threshold figures are difficult to assess and the graphs are far too busy (especially E and F). It is curious that the wild-type who are 44 weeks old do not show more of a hearing loss since they are C57 mice (known to show early age induced hearing loss).

The scanning electron microscopy figure showing the stereocilia is also of poor quality (9 kHz region for the inner hair cells from the WT looks rather pathological). There is no need to show these figures since there is only a snap shot from two narrow regions and there are no differences. The western blots are of poor quality and should be improved.

The mRNA analysis included one gene for internal normalization. Apparently no tests were performed to show that this was the most stable when comparing different samples over different ages. According to figure 1 the *Hprt* is not the best internal control for the mutant (Fig. 1C).

Moreover, why didn't the authors use control proteins targeted to post-synaptic proteins?

The histological section of the cochlea in Figure 1 is of too low magnification. It is not possible to evaluate the inner or outer hair cells in these micrographs.

The quality of the immunocytochemistry and confocal images at times over-worked such that the contrast and brightness is too exaggerated.

Summary:

The study is of interest but is very unfocussed. The quality of the figures needs to be improved.

** As a service to authors, EMBO provides authors with the possibility to transfer a manuscript that one journal cannot offer to publish to another EMBO publication. The full manuscript and if applicable, reviewers reports are automatically sent to the receiving journal to allow for fast handling and a prompt decision on your manuscript. For more details of this service, and to transfer your manuscript to another EMBO title please click on Link Not Available.**

Transfer Date

09 June 2015

2nd Editorial Decision

11 June 2015

Thank you for transferring your study to EMBO Molecular Medicine.

I have now read the study and referees comments attentively and discussed them together with my colleague. We certainly agree that the study is of interest for EMBO Molecular Medicine. We would like to invite you to submit a revised version of your article, particularly focusing on conclusively reporting the hearing loss phenotype in mice and adding clinical insights.

While molecular mechanism is not critical for us, we still would expect convincing data explaining progressive hearing loss in these mice. We would like to encourage you to address comments from both referees keeping conclusiveness in mind, without performing further reaching additional experiments (if in doubts, you are more than welcome to contact me for further discussion regarding what you can or cannot do within {plus minus} 3-months).

Unlike referee 2 suggesting to remove the patient data, we would like you to expand on this by, for example, providing audiometric data and any other sort of data that would help link the phenotype in mice to the patient phenotype (that must be described in more details, along with ethical statements for using this patient's personal information that is missing from the article). The translational aspect of the work is equally a point that could be strengthened, and you may consider treating mice with oestrogens for example as suggested.

We will send the paper back to referees, for this reason please make sure to provide a full rebuttal, detailing experiments performed or text modifications.

Please note that it is EMBO Molecular Medicine policy to allow only a single round of revision and that, as acceptance or rejection of the manuscript will depend on another round of review, your responses should be as complete as possible.

Please read below for important editorial formatting.

I look forward to receiving your revised manuscript.

1st Revision - authors' response

11 November 2015

We now submit our revised manuscript by Buniello *et al.* with the new title "Wbp2 is required for normal glutamatergic synapses in the cochlea and is crucial for hearing" in which we describe the finding of a new gene involved in deafness in both humans and mouse. We use the mouse to investigate the underlying mechanisms of hearing loss and find evidence of involvement of estrogen/progesterone signalling in the pathology. The earliest defect we see is swelling of the nerve endings below inner hair cells, a sign of glutamate excitotoxicity. We believe that these findings will open opportunities for new therapeutic approaches.

Our responses to the reviewers' comments are listed at the end of this letter. In responding, we have been able to add significant new data, in particular the finding of a second deaf child with two *WBP2* mutations, but also new analysis of *Wbp2* isoforms in the cochlea, new ultrastructural analysis, and Distortion Product Otoacoustic Emission recordings to interrogate outer hair cell function.

As you requested, we are uploading a version of our manuscript with all changes accepted. The line numbers in the responses that follow all refer to the new version.

We hope you will find these are important and interesting findings in a field that has not seen much progress in developing medical treatments.

Referee #1:

The study by Buniello et al reports that loss of Wbp2 function in mice results in hearing loss and the identification of a child with compound heterozygosity in the Wbp2 gene that presents with profound deafness. Furthermore, using physiological histological analysis, they go on to show that the inner ear of Wbp2 mutant mice shows a progressive disruption of the postsynaptic terminals innervating inner hair cells. Moreover, they document that KO of Wbp2 results in dysregulation of estrogen related and post-synaptic genes. Based on these observations, the authors conclude that Wbp2 is a critical player in hearing and propose that the synaptic defects result from excitotoxic events downstream of the increased expression of post-synaptic molecules such as PSD95 and Shank3.

The experiments are well done and the data provides a strong link between Wbp2 and progressive hearing loss in mice. However, many of the claims and conclusions are not supported by the data. Furthermore, the relevance of Wbp2 mutations for hearing loss in humans is tenuous.

Response: We have now added details of a second deaf child with two *WBP2* mutations, boosting the link between *WBP2* and deafness.

Major comments:

1. The paper does not really explain how the KO mice were generated. Specifically, the paper provides a depiction of the construct (Fig 1A), but I could not find any place where they say if exon 2 was knocked out by Cre-mediated recombination and how this have been done, neither if they phenotypic analysis was done in mice carrying the LacZ/neo cassette or if those sequences were flipped out.

Response: Figure 1A presents the new *Wbp2* allele, not the construct. We have changed the legend of Fig 1A (line 745) to clarify that this represents the allele to avoid confusion about whether it is the targeting construct or the allele, and we have added citations to the two references that describe the targeting and production of the mouse to the first sentence of the results (line 93). In the methods section, we give the details of the mouse allele and present a diagram illustrating the design of the allele (Fig1A). We state in the original version “Wbp2-deficient (*Wbp2*^{tm2a(EUCOMM)Wtsi}) mice were produced at the Wellcome Trust Sanger Institute and carry a knockout-first conditional-ready allele (Skarnes, 2011; White et al, 2013) in which a promoterless cassette including *LacZ* and *neo* genes were inserted in intron 1-2 of the *Wbp2* gene located on chromosome 11 (Fig1A).” (lines 431-434) Further details of how the targetting was carried out and the mice generated have been published in the two references cited, and the ES cell targeting and blastocyst microinjection are standard techniques so we have not repeated the description here. We did not knock out exon 2, and we suggest that it should not be necessary to say what we did **not** do in the text. If we had knocked out this exon, we would have said so. The mice used were carrying the allele *Wbp2*^{tm2a(EUCOMM)Wtsi} as stated in the methods.

2. It is surprising that the authors did not test outer hair cell function by DPOAEs in these mice. This is essential to understand the hearing phenotype.

Response: Yes we agree with this point. Since submitting this manuscript, we have recorded DPOAEs in Wbp2-deficient mice and littermate controls at 4 and 21 weeks of age. Results and methods are shown in Figure 2H,I and see lines 117-121; 293-294; 307-309; 480-494 in the revised manuscript. We added the following new paragraph describing the results: “DPOAEs were recorded and showed raised thresholds, similar to that seen in ABR measurements. In mice aged 4 weeks old, 2f1-f2 DPOAE thresholds for 6-18kHz f2 tones were comparable in mutants and littermate controls, but were elevated for 24 & 30kHz f2 tones. Impairment of DPOAEs was progressive. In 21 week old mutants, 2f1-f2 DPOAE thresholds for all test frequencies were elevated, particularly at 18-30kHz.” Furthermore, we have extended our report of our examination of OHCs in the confocal experiments, lines 219-220 and Fig 5A,C.

The new finding of raised thresholds for DPOAEs suggests that our original suggestion of a primary post-synaptic pathology in *Wbp2* mutant cochleas needs to be modified, because abnormal DPOAEs are an indicator of outer hair cell dysfunction. Therefore, we have changed the discussion of our findings to alter the balance, on lines 51-54; 353-354; 368-371

3. While the evidence that *Wbp2* LOF results in progressive loss in mice is compelling, the mechanistic hypothesis about how this occurs are based purely in correlations that are intriguing but not solid or defensible.

Response: We have presented a significant amount of evidence that supports the hypothesis that *Wbp2* is involved in post-synaptic function of IHCs via estrogen and/or progesterone signalling, but we agree that we would need to carry out further experiments to demonstrate this mechanism conclusively. Therefore, we have moderated our conclusions in the discussion on lines 32-34; 270; 275-277; 279-280; 376, and refer to synaptic defects rather than post-synaptic defects elsewhere.

4. The authors claim that the phenotype in the *Wbp2* mutant mice resembles cochlear excitotoxicity observed after noise exposure or ischemia, with common pathological hallmark of cochlear nerve terminal swelling. However, the authors observed a normal juxtaposition between *ctbp2* and *psd-95* (Fig7C), suggesting that the nerve terminal is still in contact with the presynaptic terminals. This is an intriguing observation considering the massive terminal swelling in the mutants. To understand the pathology it will be necessary to interrogate the structural features of the synaptic terminals using EM.

Response: We agree that transmission EM would be useful to understand better the pathology we observe in the *Wbp2* mutant cochleas. Therefore, we have now examined samples using TEM, as shown in revised Figures 5 and 6, and described the observations in the manuscript on lines 220-225; 235-239; 275-277; 300; 552-558. TEM showed that pre and post-synaptic densities are often broadly aligned in the mutant cochlea at 4 weeks of age, even when the contact involves a swollen terminal (see Fig6 A-G).

5. The data on the summing potential (SP) (Fig 2I) has very large error bars, making it hard to understand how it is significantly different. Similarly, the images used to claim alterations in inner hair cell ribbon distribution are not convincing. 3D reconstructions might help make the point more clear and compelling.

Response: Fig 2I shows data on the wave 1 amplitude, not SP. We have now moved these plots to an expanded view figure (see Figure EV2) and have re-plotted the data in a clearer way to avoid confusions in Figure EV2 D-I. We carried out appropriate statistics on the data (ANOVA, 2 Way, with Holm-Sidak multiple comparisons, $p < 0.05$).

Regarding the hair cell ribbon distribution, we agree with the reviewer that the images are not convincing. This is a subtle point that is difficult to demonstrate adequately in a figure, even in 3D, so we have withdrawn this observation. Deleted from lines 216 & 230.

6. In animal models, complete *wbp2* knockout results in progressive hearing loss that is mild at young. However, in the 5-yr human subject, the two *wbp2* alleles associate with profound childhood deafness. It is not sufficiently convincing that the two *wbp2* mutations could explain the disease phenotype. The authors should explain this concern. In addition, unlike what the authors described: "The A224V mutation is novel and predicted to affect protein function (SIFT; score=0.00)", SIFT analysis with ENSP number (ENSP00000254806) predicts A224V to be "TOLERATED" for protein function, with SIFT score of 0.3.

Response: As the mouse and human have very different lifespans, it is difficult to align rates of progression of hearing loss in the two species. Furthermore, we have only a single timepoint with audiometric data from the human and cannot comment on whether there was normal hearing at an earlier stage. It is known that there is a significant amount of postnatal onset and progression of hearing loss in the first decade of life in children (Fortnum et al. 2001, BMJ 323:1-6) so we cannot presume the level of auditory function at birth from a single audiogram obtained later in childhood.

Since the paper was submitted, we have identified another deaf child carrying two mutations in heterozygosis in the *WBP2* gene (Fig3 and lines 87-89; 131-155; 317-333; 363-364; 517-528). The two deaf probands share one mutation in *WBP2* exon 5 and one child has both mutations in exon 5, so we have investigated the possibility that exon 5 may be particularly important in hearing (Fig 4; lines 157-177; 322-333; 530-539). We found that the organ of Corti only expressed the long isoform including exon 5, and the short isoform that omits exon 5 was barely detectable. In contrast, brain

tissue expresses both isoforms with and without exon 5 so may not be so dependent upon exon 5. Therefore, the ear may be especially sensitive to mutations in exon 5 as it does not appear to use the shorter isoform. These observations are presented in Fig 4.

With respect to the SIFT analysis, SIFT offers several different input options, and the results differ depending on what is used. One option is to input the human protein ID and the amino acid substitution, and the outcome of that query for A224V is indeed “tolerated”, with a score of 0.3, as noted by the reviewer. However, inputting just the protein ID does not appear to take into account the multiple Wbp2 transcripts. Using the Ensembl Variant Predictor tool, which checks multiple transcripts, the scores for A224V range from 0.07 in transcript ENST00000433525, to 0.05 in transcript ENST00000585462, 0.04 in transcripts ENST00000254806, ENST00000591399 and ENST00000590221, and 0.03 in transcript ENST00000586257 (the transcripts which contain the mutant amino acid; transcripts ENST00000626827, ENST00000593002, ENST00000589642 and ENST00000587374 don't). The score of 0.07 is considered tolerated, but all the others are classed as deleterious. For all those transcripts, the PolyPhen2 prediction is “possibly damaging”. If the SIFT predictions are correct, then the impact of the mutation will depend on which transcript is expressed in the inner ear.

When we ran the SIFT analysis reported in our manuscript, we chose a different option, which permitted us to choose the orthologues against which the sequence would be compared (SIFT Sequence, <http://sift.jcvi.org/>). Our sequences for comparison at that amino acid were gorilla, mouse, rat, anole lizard, dolphin, chimpanzee, zebrafish, armadillo, chicken, blowfish and tarsier, all of which have the “A” amino acid. Only the Xenopus sequence was different, lacking the 8 amino acids in positions 220 to 228 (numbered with respect to the human sequence), so it is unsurprising that the SIFT prediction, which is based on conservation, is that the substitution would affect the protein function.

We have since tested the A224V mutation with four other effect predictors: Fathmm, Mutation Assessor, Mutation Taster and PhD-SNP. Fathmm predicts the mutation is tolerated, Mutation Assessor scores it as “Medium”, Mutation Taster predicts it to be disease-causing and PhD-SNP to be Neutral.

See new Table I and lines 146-155; 157-177; 322-333; 517-528 for the new analysis.

7. It would be important to test if the interaction between Wbp2 LOF and changes in expression of the other genes (Psd-95, Shank3, Esr1, Esr2 and Pgr) are specific to the inner ear or are relevant to other parts of the nervous system, and determine the reason for the specificity if it exists. If not, it would be important to explain why other excitatory synapses are not affected.

Response: We agree, but we were not able to collect enough new samples to obtain robust data within the timescale available. We plan to look at this question in future experiments when more samples are available.

Referee #2:

General Comments:

1) This manuscript describes peripheral auditory changes in Wbp2-deficient mice. The authors have performed numerous assays to characterize the peripheral auditory organ in these mice. While the careful characterization of this mouse mutant is commendable, it tends to distract from the main point of the study. Many of these assays showed no differences between the mutants and the wild-type mice and these finding should be placed in the supplementary section.

Response: We have moved further non-essential data to the expanded view figures as suggested. However, some negative findings are critical to the interpretation of the pathology (such as the single IHC recordings and lack of other pathological features in both mouse and humans) so we have retained some of these results in the main text, abbreviated where possible. See Figures EV1, EV2, EV3, lines 122-126; 179-204..

2) The section with the genetic mutation in the 5-year old boy is interesting, but without any audiometric tests or other types of characterization this data seems to hang in the air and should be removed from the manuscript.

Response: The boy was recorded as having profound prelingual deafness but no audiometry was available. Since submitting the manuscript, we have found a second child with two *WBP2* mutations and available audiometry, which is now presented in revised figure 3. The finding of two unrelated children with hearing impairment associated with *WBP2* mutations and the mouse mutant with hearing impairment due to *Wbp2* mutation amounts to strong evidence that *WBP2* is involved in deafness and should be considered as a candidate gene in other cases of human hearing loss. Fig 3 and described on lines 131-155.

3) The sections of the manuscript where there are differences between the two groups are not well performed or presented. It is not understood why the authors have selected two narrow regions to study the post-synaptic changes (i.e. the main part of the study). The authors concentrate on 9 and 24 kHz that corresponds to the apex and towards the base of the cochlea. Quantification should have been performed throughout the entire cochlea for the data showing differences between the two groups. How can the authors be sure that there are no changes in the other regions of the cochlea? This is a weak aspect to the study.

Response: We have analysed the organ of Corti throughout the whole length of the cochlea, but we needed to select specific regions to present in the figures so we chose to report the data on the 9 and 24 kHz best-frequency regions to make the paper more focused. We measured the percentage distance along the length of the cochlear duct from base to apex and used the frequency-place map of Müller *et al* (2010) to determine the regions to quantify. These observations are representative of the general gradient in the reported features from base to apex. We have clarified our approach now in the revised methods section (lines 212-214; 618-620) Moreover, we have added a third point (16kHz) in the quantification of the pre-synaptic ribbons at 4 weeks (Fig 5B).

Specific Comments:

1) The ABR threshold figures are difficult to assess and the graphs are far too busy (especially E and F). It is curious that the wild-type who are 44 weeks old do not show more of a hearing loss since they are C57 mice (known to show early age induced hearing loss).

Response: We have simplified the plots of ABR thresholds and removed some of the panels to an expanded view figure. However, we kept the pale grey plots showing individual mutants' thresholds because we think it is a better indicator of the range of data than showing the means \pm SD alone, and is recommended by statisticians (Festing & Altman 2002, ILAR Journal 43:244-258).

We note the comment about C57BL/6 mice and the reports of early-onset threshold increases in this strain. It is difficult to compare thresholds of mice across different laboratories. We take care to minimise the exposure to noise in the mouse holding rooms (no radios allowed for example) and hold the mice in individually ventilated cages with calibrated background noise levels. The mice we study are the C57BL/6N substrain, different to other reports which use C57BL/6J. We only measure hearing in mice that have been bred in house, while other studies used mice that were shipped from a supplier with uncontrolled exposure to noise as well as other environmental stressors. Any of these factors, or others, may contribute to the relatively good preservation of hearing we find in our wildtype control mice.

2) The scanning electron microscopy figure showing the stereocilia is also of poor quality (9 kHz region for the inner hair cells from the WT looks rather pathological). There is no need to show these figures since there is only a snap shot from two narrow regions and there are no differences.

Response: We agree that these are not ideal preparations, but the mutants and wildtypes were processed at the same time in an identical manner so we should be able to distinguish any significant differences between genotypes despite the artefactual splaying of IHC stereocilia bundles. We have moved the SEM figures to an Expanded View figure. See Figure EV2A.

3) The western blots are of poor quality and should be improved.

Response: We agree that the inner ear westerns are not good quality, but this may result from the very limited amount of tissue available from the mouse. We have replaced the *Wbp2* western

originally in Fig1B with a clearer one obtained from brain tissue. We also removed the GluR2/3 western (previous Fig6B), as it is redundant with the immunolabelling.

4) The mRNA analysis included one gene for internal normalization. Apparently no tests were performed to show that this was the most stable when comparing different samples over different ages. According to figure1 the Hprt is not the best internal control for the mutant (Fig. 1C). Moreover, why didn't the authors use control proteins targeted to post-synaptic proteins?

Response: The Hprt levels were more variable than we would hope for, but the amount of tissue we have available from the organ of Corti or cochlea is very limited which makes these measurements challenging. In previous tests using different molecules as calibrators, including Gapdh, p75, Jag1 and Hprt, performed on cochlear tissues, we established that Hprt is the most reliable of these calibrators for the ages examined. cDNA obtained from the cochlea is a very limited resource, so we do not generally use multiple calibrators. We did not use control proteins from among known post-synaptic proteins because they could all show impaired expression and so were the subject of our investigation rather than independent controls. Irrespective of the calibrator used, we find that Wbp2 is strongly down-regulated in the mutants' ears and eyes (Fig 1C). We have now replotted the graph in Figure 1C in a clearer way.

5) The histological section of the cochlea in Figure 1 is of too low magnification. It is not possible to evaluate the inner or outer hair cells in these micrographs.

Response: We have replaced the panel with one of higher magnification, see Fig 1F.

6) The quality of the immunocytochemistry and confocal images at times over-worked such that the contrast and brightness is too exaggerated.

Response: We have adjusted the brightness and quality of the confocal images in Figs 5, 6 and 7. The green signal after neurofilament labelling is always very strong in the *Wbp2* mutants, because of the enlarged afferent nerve terminals. (Fig 5A).

Additional Author correspondence

13 December 2015

We were pleased to hear that one of the reviewers was supportive of publication. The other reviewer appears to have added a new suggestion for further experiments, which were not requested by any of the previous four reviewers including him/her.

Unfortunately we do not have any in vitro data to support the human mutations. My laboratory is not set up for in vitro screens as we generally look at function in vivo, so we would need to find someone to collaborate with us to carry out these tests. Even if we started today, it would take between 6 and 12 months to establish and validate a suitable screen for Wbp2 function and test the three human mutations.

We have added considerable additional data since our first submission, but it is always possible for reviewers to think of extra experiments for authors to carry out - there are endless extra experiments that could be carried out to strengthen any study. It is really disappointing to hear that reviewer 1 has now, at this late stage, asked for additional experiments that they did not suggest earlier. Why did they not suggest these in their first review? Two of the three human mutations were reported in our first submission, so human mutations are not completely new data in our revision.

We have found three different human mutations in two unrelated children with no evidence of mutations in any other known deafness gene plus one mouse mutation, all associated with deafness, which seems to be strong evidence for a role for Wbp2 in auditory function.

I look forwards to hearing from you, and assure you that any rewriting necessary will be carried out with urgency and the manuscript returned to you before Christmas.

Thank you for the submission of your revised manuscript to EMBO Molecular Medicine and for replying in details to my last letter. We have now received the enclosed reports from the referees who were asked to re-assess it. As you will see, although one reviewer is fully supportive while one remained not entirely satisfied, I am pleased to inform you that we will be able to accept your manuscript pending the following final amendments:

1) please thoroughly address all remaining criticisms from both referees in writing in the main text and in a rebuttal letter.

Please submit your revised manuscript within two weeks. I look forward to seeing a revised form of your manuscript as soon as possible.

***** Reviewer's comments *****

Referee #1 (Remarks):

In response to the initial review, the authors have made a number of new experiments that resulted in significant new data, including TEM, measurements of DPOAEs and a new human case. The new information has led to significant changes to the paper and the interpretation of the phenotypes. While the story is a bit more complete, I think that the data does not provide compelling insights into the roles of Wbp2 in the inner and in deafness, or on ways this knowledge could be used to develop therapeutic strategies to treat hearing loss.

Specific comments:

1. As the title indicates, the authors continue to put emphasis on the contribution of inner hair cell glutamatergic synapse disruption to the hearing loss caused by Wbp2 mutations. However, in light of the new DPOAE data (Fig 2H,I), it seems that the auditory phenotype could well be primarily caused by outer hair cell dysfunction. In addition, the authors also observed changes in cochlear SP (Fig 2G and Fig EV1A-C) and detachment hair cell ribbons (Fig 6B-G), suggesting that the inner hair cells are also affected. Although the wbp2 mutants show remarkable swelling of the auditory nerve terminals, it is unclear how this change may be related to the ABR and DPOAE threshold shifts in wbp2 mutant mice.
2. Although the authors suggest that wbp2 compound heterozygous mutations are associated with human deafness cases, it is unclear if these mutations indeed impair wbp2 function. In vitro cell assay demonstrating the effects of these mutations on wbp2 expression and activity will be essential.
3. Lastly, the paper does not provide clear translational insights.

Referee #3 (Comments on Novelty/Model System):

The mouse is an excellent model system for studying human hereditary hearing loss.

Referee #3 (Remarks):

The description of Wbp2 as a gene involved in mouse and human deafness, and the introduction of estrogen and progesterone receptors in this process is compelling. The experiments support the conclusions well. All revisions from prior version add critical components to the paper. The medical relevance is important for hearing and deafness.

Minor but essential revisions for human variants:

Follow guidelines for human nomenclature
<http://www.genenames.org/about/guidelines>

Change 'mutations' to 'variants' or 'pathogenic variants.'

For example, "Two of the three human mutations.." should be changed to "Two of the three human

variants...."

First time proteins referred to, add parentheses, since they are predicted
 c.671C>T; p.(A224V)
 c.478G>A:p.(A160T)
 Thereafter not necessary

Use accepted nomenclature for variants:

A160T
 Change to Ala160Thr

M163L
 Change to Met163Leu

2nd Revision - authors' response

18 December 2015

Referee #1:

In response to the initial review, the authors have made a number of new experiments that resulted in significant new data, including TEM, measurements of DPOAEs and a new human case. The new information has led to significant changes to the paper and the interpretation of the phenotypes. While the story is a bit more complete, I think that the data does not provide compelling insights into the roles of Wbp2 in the inner and in deafness, or on ways this knowledge could be used to develop therapeutic strategies to treat hearing loss.

Specific comments:

1. As the title indicates, the authors continue to put emphasis on the contribution of inner hair cell glutamatergic synapse disruption to the hearing loss caused by Wbp2 mutations. However, in light of the new DPOAE data (Fig 2H,I), it seems that the auditory phenotype could well be primarily caused by outer hair cell dysfunction. In addition, the authors also observed changes in cochlear SP (Fig 2G and Fig EV1A-C) and detachment hair cell ribbons (Fig 6B-G), suggesting that the inner hair cells are also affected.

The title refers only to glutamatergic synapses, not to inner hair cell synapses. We have demonstrated defects in cochlear synapses of both inner and outer hair cells, so we do not think the title is misleading.

We agree that our evidence does indicate that inner hair cells are affected by the mutation, but it is not clear whether these are primary defects or secondary changes following a primary post-synaptic defect, or both. The nerve ending swelling is the first abnormal feature we detect during development and maturation. We have discussed this in the discussion, lines 298-313, and concluded that the weight of evidence suggests there is a primary post-synaptic defect but that a primary inner hair cell defect cannot be ruled out. We have adjusted the text here to make this clearer as follows.

Lines 298-299: "The most likely site of the initial IHC dysfunction is the post-synaptic region of the afferent synapse rather than the pre-synaptic complex, for a number of reasons."

Lines 308-311: "However, there are some signs of pre-synaptic defects too, including the reduced SP amplitude suggesting impaired depolarisation of mutant IHCs, and some misplaced ribbons observed by TEM, so we cannot rule out a direct effect of the mutation on IHCs."

However, we find it difficult to envisage how a primary outer hair cell defect could cause the IHC synaptic defects that we observe.

Although the wbp2 mutants show remarkable swelling of the auditory nerve terminals, it is unclear how this change may be related to the ABR and DPOAE threshold shifts in wbp2 mutant mice.

This relationship is usually assumed to be a causal relationship in the literature, but we agree that it is a strong association rather than definitively proven to be a causal relationship. The best evidence that we know of to support the proposal that the swelling causes an increase in ABR and DPOAE

thresholds is a classic paper by Don Robertson, which we have cited and discussed in the text on lines 385-392. Robertson used noise exposure to align swelling of nerve endings with impaired single unit responses: "Exposures which caused swelling produced a loss in sensitivity of the flat low frequency 'tail' of the frequency-threshold curves of single auditory neurons whose most sensitive frequency was a 1/2 octave higher than the exposure frequency. The findings are consistent with the notion that dendritic swelling causes a non-selective decrease in sensitivity to all frequencies of sound."

2. Although the authors suggest that wbp2 compound heterozygous mutations are associated with human deafness cases, it is unclear if these mutations indeed impair wbp2 function. In vitro cell assay demonstrating the effects of these mutations on wbp2 expression and activity will be essential.

These will be interesting and useful experiments to do in the future, but we have not been able to carry out these in vitro assays in the timescale allowed. In the meantime, we have presented three human mutations associated with deafness in two children with no other predicted pathogenic mutations in genes previously known to be involved in deafness, plus one mouse mutation with a clearcut effect on auditory function. We suggest that this is strong evidence supporting a role for Wbp2 in deafness.

3. Lastly, the paper does not provide clear translational insights.

We do not claim that Wbp2 itself is a good direct target for drug discovery, but our finding does suggest that further exploration of the regulatory pathways and functional interactions involving Wbp2 may present clear targets. If Wbp2 expression can be upregulated, this may protect hair cell synapses from excitotoxicity while avoiding unwanted side effects, because Wbp2 deficiency appears to have a specific influence on hearing without leading to any other detected abnormalities. We suggest that this does provide a clue to a potential therapeutic route.

Referee #3:

The description of Wbp2 as a gene involved in mouse and human deafness, and the introduction of estrogen and progesterone receptors in this process is compelling. The experiments support the conclusions well. All revisions from prior version add critical components to the paper. The medical relevance is important for hearing and deafness.

Minor but essential revisions for human variants:

Follow guidelines for human nomenclature

<http://www.genenames.org/about/guidelines>

Change 'mutations' to 'variants' or 'pathogenic variants.'

For example, "Two of the three human mutations.." should be changed to "Two of the three human variants...."

We have changed the word mutation to variant throughout the text as requested.

First time proteins referred to, add parentheses, since they are predicted

c.671C>T; p.(A224V)

c.478G>A;p.(A160T)

Thereafter not necessary

These parentheses have been added

Use accepted nomenclature for variants:

A160T

Change to Ala160Thr

We thought that the recommended nomenclature was the single-letter amino acid code, but we have changed the description to the three-letter version throughout the text as requested.

M163L
Change to Met163Leu

This has been changed as requested.